# DKDR: Dynamic Knowledge Distillation for Reliability in Federated Learning

**Yueyang Yuan**[1,2†], **Wenke Huang**[1,2†], **Guancheng Wan**[1†],
**Kaiqi Guan**[1], **He Li**[1], **Mang Ye**[1*]

[1] National Engineering Research Center for Multimedia Software, Institute of Artificial Intelligence,
Hubei Key Laboratory of Multimedia and Network Communication Engineering,
School of Computer Science, Wuhan University, Wuhan, China.
[2] Guangdong Laboratory of Artificial Intelligence and Digital Economy (SZ)
{yueyangyuan, wenkehuang, guanchengwan, yemang}@whu.edu.cn

## Abstract

Federated Learning (FL) has demonstrated a promising future in privacy-friendly collaboration but it faces the data heterogeneity problem. Knowledge Distillation (KD) can serve as an effective method to address this issue. However, challenges arise from the unreliability of existing distillation methods in multi-domain scenarios. Prevalent distillation solutions primarily aim to fit the distributions of the global model directly by minimizing forward Kullback-Leibler divergence (KLD). This results in significant bias when the outputs of the global model are multi-peaked, which indicates **the unreliability of distillation pathway**. Meanwhile, cross-domain update conflicts can notably reduce the accuracy of the global model (teacher model) in certain domains, reflecting **the unreliability of the teacher model** in these domains. In this work, we propose **DKDR** (**D**ynamic **K**nowledge **D**istillation for **R**eliability in Federated Learning), which dynamically assigns weights to forward and reverse KLD based on knowledge discrepancies. This enables clients to fit the outputs from the teacher precisely. Moreover, we use knowledge decoupling to identify domain experts, thus clients can acquire reliable domain knowledge from experts. Empirical results from single-domain and multi-domain image classification tasks demonstrate the effectiveness of the proposed method and the efficiency of its key modules. The code is available at https://github.com/YueyangYuan/DKDR.

## 1 Introduction

Federated learning is a collaborative paradigm [21, 60, 27, 14–16, 62], enabling multiple clients to jointly train a shared global model [39, 28, 15] while ensuring privacy protection [52]. However, the distributed data is collected from different sources with diverse preferences and brings the non-independent and identically distributed (non-IID) characteristics. Knowledge distillation [13, 4] addresses this challenge effectively by aligning the outputs of local models with the global model. It brings the optimization objectives of each client closer together thus resolving the problem.

However, existing distillation methods [25, 11, 36, 5] typically use forward KLD to fit the distributions of the global model. We argue that this approach is unreliable in multi-domain scenarios. Given the global model distribution $Z(y|x)$ and the local model distribution $Z_w(y|x)$ parameterized by $w$, standard knowledge distillation objectives aim to minimize the forward KLD between them, denoted as $KL[Z|Z_w]$. This approach compels $Z_w$ to encompass all modes of $Z$. However, we

---

[†] Equal Contribution.
[*] Corresponding Author.

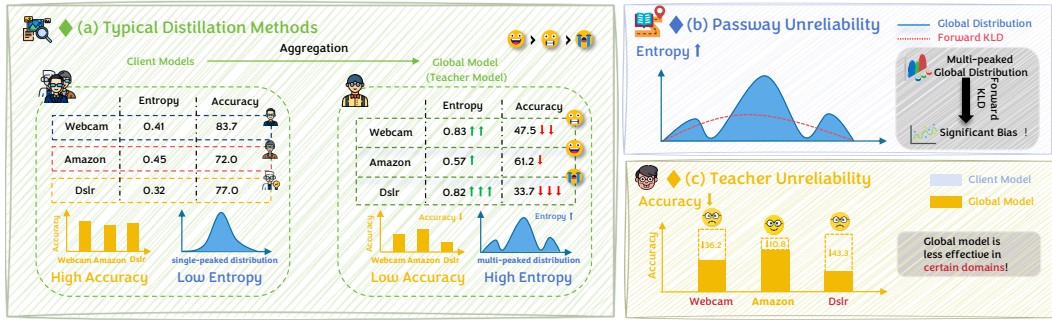

Figure 1: **Problem illustration. (a) Typical Distillation Methods** exist two unreliability problems as follows. **(b) Pathway Unreliability**: in multi-domain scenarios, the aggregated global model shows a significantly higher entropy compared to local models in the distributions over private datasets, exhibiting a multi-peak structure. In such case, minimizing forward KLD leads to significant bias; **(c) Teacher Unreliability**: after aggregation, the global model experiences catastrophic accuracy drops in certain domains. Thus in these areas, the global model is unable to serve as an effective teacher to provide high-quality guidance for clients.

notice that the global model will produce multi-peaked outputs in domains with limited data, as shown in Fig.1. In such situations, minimizing the forward KLD causes $Z_w$ to assign unreasonably high probabilities to regions of low density in $Z$ [38]. **Therefore, distillation on these domains will introduce significant bias, which is the unreliability of the distillation pathway**. This context naturally raises the following critical question: **I)** *how can we establish a reliable distillation pathway in federation?* Meanwhile, conflicts in update directions across different domains can significantly reduce the accuracy of the global model on some domains. **Thus, the global model is inherently less effective on these domains, which is the unreliability of the teacher model**. This situation prompts another intriguing question: **II)** *how can we get a reliable teacher for distillation?*

To address these challenges, we propose DKDR (**D**ynamic **K**nowledge **D**istillation for **R**eliability in Federated Learning). Concerning the issue of pathway unreliability mentioned in **I)**, we initially conduct a theoretical analysis of federated knowledge distillation (see Sec.3.2). The reverse KLD, denoted as $KL[Z_w||Z]$, is widely used in knowledge distillation. Reverse KLD promotes a mode-seeking behavior, leading $Z_w$ to focus on a singular mode of $Z$ [3, 53, 22], while forward KLD induces a mean-seeking behavior, encouraging $Z_w$ to capture the overall distribution of $Z$. Therefore, when distilling the multi-peaked distributions, forward KLD will assign high probabilities to low-density regions. In contrast, the reverse KLD prioritizes confidence intervals. Both methods exhibit different types of bias. An intuitive idea is utilizing their characteristics to design a dynamic weighting method, aiming to minimize the bias introduced by distillation pathway as much as possible. We demonstrate from both experimental and theoretical perspectives that the forward KLD prioritizes fitting the dominant regions of the global distribution, while the reverse KLD prioritizes fitting the lower-probability segments. Based on their characteristics, we introduce **Dynamic Distillation**: dynamically allocating weights to forward and reverse KLD based on the knowledge discrepancies between the Dominant Knowledge Components (DKCs) and Ancillary Knowledge Components (AKCs) (see Sec.3.2). Thus clients can precisely fit the distributions of the teacher model.

In the second place, to get a reliable teacher mentioned in **II)**, we propose **Knowledge Decoupling** to get domain experts: we first modularize knowledge into shared and unique components and then use SVD to extract the main components of unique knowledge. Subsequently, clustering techniques are employed on the refined components to identify domain experts that specialize in distinct domains. In federation, clients learn from domain experts rather than the global model, thus gaining reliable domain knowledge. Experimental results reveal that our method consistently achieves better performance than others. The main contributions are summarized as:

❶ *Re-examining KD in FL from a Reliability Perspective.* Our findings indicate that existing federated distillation methods are unreliable in multi-domain scenarios, which results in significant distillation bias and less effective guidance for clients in domains with limited data.

❷ *Novel Dynamic Multi-experts Distillation Framework for Reliability.* Building on the phenomenon of unreliable KD in FL, we effectively mitigate distillation bias and comprehensively improve the performance across domains by addressing the pathway unreliability and teacher unreliability.

❸ *Theoretical Guarantees and Experimental Validation.* We provide theoretical guarantees for our framework, and further demonstrate the effectiveness of it through comprehensive experiments.

## 2 Related Work

### 2.1 Heterogeneous Federated Learning

A pioneering work proposed the currently most widely used algorithm, FedAvg [39]. However, it suffers from performance deterioration when applied to non-i.i.d data (data heterogeneity). Shortly thereafter, a substantial body of research [29, 46, 48, 28] emerged, focusing on non-i.i.d data. These methods primarily address label distribution skew, where non-i.i.d data [18] is created by partitioning existing data based on label space with limited domain shift. FedProx [29], FedCurv [47], pFedME [48], and FedDyn [2] calculate global parameter stiffness to control discrepancies. Besides, MOON[28], FedUFO [64], FedProto[49], and FedProc[42] maximize feature-level alignment of local model and global model. Moreover, SCAFFOLD [19] and FedDC [9] leverage global gradient calibration to control local drift. Nevertheless, when private data is sampled from different data domains, these works do not consider inter-domain performance, concentrating instead on learning an internal model. Recent studies have explored related issues in unsupervised domain adaptation for target domains [43, 30] and domain generalization on unseen domains [34]. However, collecting data in the target domain can be time-consuming and impractical, while considering performance on unknown domains represents an idealistic scenario. In more realistic settings, participants are likely to be more concerned with performance across other domains, as this could directly enhance economic benefits. Our method leverage the **Knowledge Decoupling** to capture domain-specific signals and identify domain experts. It focuses on improving performance in outer domains during distillation, learning a generalizable and stable global model during the federated learning process.

### 2.2 Federated Knowledge Distillation

Knowledge Distillation (KD) [13] is a technique that has been extensively studied and applied in various areas of machine learning. Currently, KD has found widespread applications in FL, which can be broadly categorized into three main areas: addressing data heterogeneity, enhancing generalization capabilities, and mitigating catastrophic forgetting [54, 66, 58, 26, 61, 57, 17]. In terms of addressing data heterogeneity, FedFTG [65] employs a data-free knowledge distillation method to fine-tune the global model, while FedDKD [31] introduces a decentralized knowledge distillation module to distill knowledge from local models. Moreover, FedUSL [6] employs a self-label reassigning method to rectify the global model predictions. Regarding the enhancement of generalization capabilities, FedX [11] utilizes a two-sided knowledge distillation approach with contrastive learning as a core component, enabling the federated system to operate without requiring clients to share any data features. Furthermore, FedMEKT [24] develops a distillation-based multimodal embedding knowledge transfer mechanism, which allows the server and clients to exchange joint multimodal embedding knowledge extracted from a multimodal proxy dataset. Finally, to address the issue of catastrophic forgetting, FedNTD [25] proposes a novel and effective algorithm, Federated Not-True Distillation, which preserves the global perspective on locally available data exclusively for the not-true classes. Additionally, CFeD [36] performs knowledge distillation on both the clients and the server to mitigate forgetting. And DFRD [35] maintains an exponential moving average copy of the generator on the server to overcome the catastrophic forgetting, using dynamic weighting and label sampling to accurately extract knowledge. It is worth noting that **all of these methods distill knowledge directly by utilizing forward KLD, resulting in significant bias in multi-domain scenarios.** In our work, we firstly introduce **Dynamic Distillation**, which dynamically weights forward and reverse KLD for different knowledge modules and establishes precise distillation.

## 3 Methodology

### 3.1 Preliminary

**Generic Federated Learning**. In general federated learning settings [39, 29, 28, 40, 41, 59, 15], there are $K$ clients (indexed by $k$) each possessing its respective private data, denoted as $D_k = \{x_i, y_i\}_{i=1}^{N_k}$, where $N_k$ represents the number of data points held by the $k^{th}$ client. The global model parameters at the beginning of the $t^{th}$ communication epoch are denoted as $w^t$. The server broadcasts these parameters to each client, assigning them as $w_k^t \leftarrow w^t$. Each client conducts local optimization and

uploads the updated parameters back to the server for weighted parameter aggregation:

$$w_k^t \leftarrow w_k^t - \eta \nabla \sum_{i \in B_k} l(w_k^t, \xi_i), \quad w^{t+1} = \sum_k \alpha_k w_k^t, \tag{1}$$

here, the $B_k$ denotes the mini-batch sampled from the private data $D_k$, $\xi$ represents the query instance, and $\eta$ indicates the local learning rate. The optimization objective is to secure a well-performing global model through the federated learning process.

**Domain shift**. There exists domain shift among private data. Specifically, for the same label space, distinctive feature distributions exists among different participants, which can be defined as:

$$\mathbb{P}_i(x|y) \neq \mathbb{P}_j(x|y) \quad \mathbb{P}_i(y) = \mathbb{P}_j(y). \tag{2}$$

**Federated Knowledge Distillation**. In typical federated knowledge distillation settings [25, 14], clients use forward KLD to distill knowledge from the global model. Specifically, the global model $w^{t-1}$ at the end of the $(t-1)^{th}$ round involves the knowledge learned from other participants. We calculate the distribution through the $k^{th}$ client model and global model of the $(t-1)^{th}$ round on private data: $Z_{i,k}^t = f(w_k^t, x_i)$ and $Z_i^{t-1} = f(w^{t-1}, x_i)$ for private data $x_i$ w.r.t its ground truth label $y_i$. The standard KD loss function of $k^{th}$ client can be formulated as:

$$\mathcal{L}_{skd}(Z_{i,k}^t, Z_i^{t-1}) = \sigma(Z_i^{t-1}) \log(\frac{\sigma(Z_i^{t-1})}{\sigma(Z_{i,k}^t)}), \tag{3}$$

where $\sigma$ denotes softmax function. The optimization objective is to mitigate the issues such as catastrophic forgetting and data heterogeneity problem.

**KD Based on reverse KLD**. KD based on minimizing reverse KLD has been widely applied in specific scenarios due to its mode-seeking characteristics [10, 20, 56]. Unlike standard KD, its distillation function can be expressed as:

$$\mathcal{L}_{rkd}(Z_{i,k}^t, Z_i^{t-1}) = \sigma(Z_{i,k}^t) \log(\frac{\sigma(Z_{i,k}^t)}{\sigma(Z_i^{t-1})}). \tag{4}$$

Research [10] suggests that due to its mode-seeking characteristics, this distillation method is more suitable for complex tasks compared to standard distillation methods.

## 3.2 Dynamic Knowledge Distillation (DKD)

**Definition 3.1. (Knowledge Modules)** *We take digits in $Z$ in descending order and then cumulatively summed until the number of selected values surpasses $\mu$, where $\mu$ is a hyperparameter and typically defined as 0.5. The selected values are defined as Dominant Knowledge Components (DKCs), while the remaining values are termed as Ancillary Knowledge Components (AKCs), formulated as:*

$$a(j) = \begin{cases} 0 & if\, z_{j,k}^t \in AKCs \\ 1 & if\, z_{j,k}^t \in DKCs \end{cases}, \tag{5a}$$

$$\min \sum_{j=0}^n a(j) \quad s.t. \quad \sum_{j=0}^n a(j) z_{j,k}^t \geq \mu. \tag{5b}$$

**Definition 3.2. (Knowledge Discrepancy)** *Knowledge discrepancy reflects the distance between two distributions. The knowledge discrepancy $\gamma_s$ within AKCs and $\gamma_l$ within DKCs are defined as:*

$$\gamma_s = \sum_{j=1}^n (1 - a(j))|z_j^{t-1} - z_{j,k}^t|, \tag{6a}$$

$$\gamma_l = \sum_{j=1}^n a(j)|z_j^{t-1} - z_{j,k}^t|. \tag{6b}$$

**Theoretical Analysis.** Forward KLD's mean-seeking characteristics result in unreliable distillation when the global model has multi-peaked distributions. Conversely, reverse KLD will also lead to distinct bias due to its mode-seeking nature. Forward KLD and reverse KLD are adept at fitting different regions of the distribution. Therefore, how to balance forward and reverse KLD to minimize distillation bias becomes a key issue. This naturally leads us to reflect on the fundamental reasons behind the different behaviors of forward and reverse KLD. Let $Z_{i,k}^t = (z_{1,k}^t, z_{2,k}^t, ..., z_{n,k}^t)$ and $Z_i^{t-1} = (z_1^{t-1}, z_2^{t-1}, ..., z_n^{t-1})$, where $n$ denotes the size of $Z$. The $\mathcal{L}_{skd}$ and $\mathcal{L}_{rkd}$ can be denoted as:

$$\mathcal{L}_{skd} = \sum_k z_j^{t-1} \log(\frac{z_j^{t-1}}{z_{j,k}^t}), \tag{7a}$$

$$\mathcal{L}_{rkd} = \sum_k z_{j,k}^t \log(\frac{z_{j,k}^t}{z_j^{t-1}}). \tag{7b}$$

The gradient for $z_{j,k}^t$ under forward and reverse KLD can be calculated by the chain rule as follows:

$$\frac{\partial \mathcal{L}_{skd}}{\partial z_{j,k}^t} = z_{j,k}^t - z_j^{t-1}, \tag{8a}$$

$$\frac{\partial \mathcal{L}_{rkd}}{\partial z_{j,k}^t} = z_{j,k}^t \log(\frac{z_{j,k}^t}{z_j^{t-1}}) - \mathcal{L}_{rkd}. \tag{8b}$$

Considering the converge condition of forward and reverse KLD:

$$\frac{\partial \mathcal{L}_{s(r)kd}}{\partial z_{j,k}^t} = 0, \forall j \in (1, 2, 3..., n), \tag{9}$$

we can infer that for both two methods, the sufficient and necessary condition for converge is:

$$z_{j,k}^t = z_j^{t-1}, \forall j \in (1, 2, 3..., n). \tag{10}$$

According to Eq.(10), both the forward and reverse KLD have the same optimal projective. Thus, the fundamental reason for their differing behaviors is the optimization process. Considering Eq.(7a), larger $z_j^{t-1}$ means a larger weight in total loss and also more likely to generate a larger $\log(z_j^{t-1}/z_{j,k}^t)$. Hence, fitting the area with larger $z_j^{t-1}$ is the priority of forward KLD. What's more, when $(z_j^{t-1}/z_{j,k}^t)$ goes to $+\infty$ the forward KLD goes to $+\infty$. Therefore, $z_{j,k}^t$ would try to cover as many peaks of $z_j^{t-1}$ as possible, leading to the mean-seeking behavior of forward KLD. Similarly, considering Eq.(7b), $(z_{j,k}^t/z_j^{t-1})$ is easier to be $+\infty$ when $z_j^{t-1}$ gets smaller, leading to a larger loss. Therefore, fitting the area with smaller $z_j^{t-1}$ is the priority of reverse KLD. It avoids $(z_j^{t-1}/z_{j,k}^t)$ go to $0^+$, which means $z_{j,k}^t$ shouldn't be too large when $z_j^{t-1}$ is small, leading to mode-seeking behavior of reverse KLD.

**Empirical Analysis.** We perform distillation using forward and reverse KLD separately, calculating the average knowledge discrepancies between the client models and the global model across the two knowledge modules (DKCs and AKCs, defined in Eq.(5)) on Cifar-100 with 10 cilents for 100 communication epochs, as illustrated in Fig.2. The experimental results indicate that when using forward KLD, the differences in the DKCs are lower, while when using reverse KLD, the differences in the AKCs are lower.

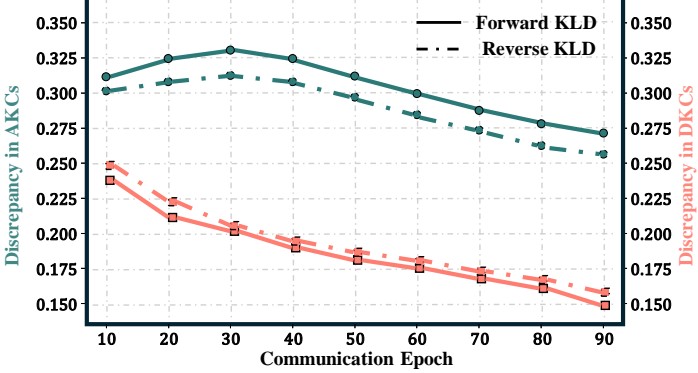

Figure 2: **Empirical Analysis.** The dashed line represents distillation based on reverse KLD, while the solid line denotes distillation based on forward KLD. For detailed information, please refer to the left main text.

**Method.** Based on these observations and analysis, it is intuitive to assign weight to the forward and the reverse KLD according to the knowledge discrepancies between the client model and the teacher model. In cases of disparity in DKCs, prioritize the forward KLD; Conversely, when there is a significant difference in AKCs, prioritize the reverse KLD. Combined with Eq.(3) and Eq.(4), the dynamic knowledge distillation loss function $\mathcal{L}_{dkd}$ is as follows:

$$\mathcal{L}_{dkd}(Z_{i,k}^t, Z_i^{t-1}) = \frac{\gamma_s}{\gamma_s + \gamma_l}\mathcal{L}_{rkd}(Z_{i,k}^t, Z_i^{t-1}) + \frac{\gamma_l}{\gamma_s + \gamma_l}\mathcal{L}_{skd}(Z_{i,k}^t, Z_i^{t-1}), \tag{11}$$

### 3.3 Knowledge Decoupling (KDP)

In prior knowledge distillation approaches, the aggregated global model has catastrophic accuracy drops, resulting in poor performance in some domains. Therefore, global model can not provide reliable guidance in these domains. We address this issue through **Knowledge Decoupling**: Decoupling

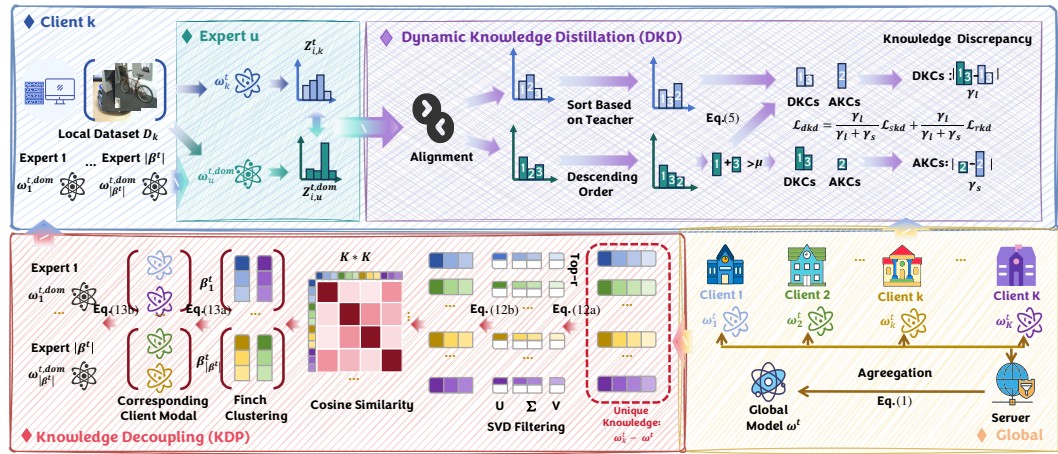

Figure 3: **Architecture illustration** of DKDR. DKDR consists of two core components: ❶ The top right box refers to Dynamic Knowledge Distillation (DKD), which adaptively weights forward and reverse KLD based on knowledge discrepancies in DKCs and AKCs (Sec.3.2). ❷ The bottom left box represents Knowledge Decoupling (KDP), where we separate shared and unique knowledge by SVD filtering and clustering to identify domain experts (Sec.3.3). Clients will distill knowledge dynamically and equally from each expert.

knowledge into shared knowledge and unique knowledge globally, and extracting domain signals from unique knowledge by SVD. Then we use Finch Clustering to get domain experts. Thus, Clients distill knowledge from these experts more efficiently.

Specifically, to get domain experts, we first examine the knowledge distillation process at a finer-grained knowledge perspective. We identify two types of critical knowledge: (1) **Shared knowledge**, which benefits **multiple domains**, and (2) **Unique knowledge**, which is useful only for a **specific domain**. In federated knowledge distillation, the mixing of shared knowledge and unique knowledge obscures domain-specific signals coming from unique knowledge. A natural idea is to separate shared knowledge and unique knowledge to clarify domain-specific signals. Therefore, during the $t^{th}$ communication epoch, we consider the global model from the $(t-1)^{th}$ communication epoch as a natural placeholder to encapsulate the shared knowledge (denoted as $w^{t-1}$). Then we calculate the difference vector for each client $w_k^t$: $v_k^t = w_k^t - w^{t-1}$. This subtraction vector preserves domain-specific signals while diminishing the interference of shared knowledge. For practical use, we apply SVD [1] to filter redundant noise and clarify domain-specific signals within unique knowledge:

$$(v_1^t, v_2^t, \ldots, v_K^t) = \mathbf{U}\boldsymbol{\Sigma}\mathbf{V}^T, \tag{12a}$$

$$(v_1^{t,S}, v_2^{t,S}, \ldots, v_K^{t,S}) = \mathbf{U}_r\boldsymbol{\Sigma}_r\mathbf{V}_r^T. \tag{12b}$$

We apply truncated SVD to $(v_1^t, v_2^t, \ldots, v_K^t)$, retaining the top $r$ singular values to extract domain-specific signals. Then we use Finch Clustering on $(v_1^{t,S}, v_2^{t,S}, ..., v_K^{t,S})$ based on cosine similarity to capture domain-specific signals in unique knowledge. Then, we replace each $v_k^S$ in each cluster with the corresponding client $k$ to obtain client clusters $\beta^t$, where each $\beta_u^t \in \beta^t$ corresponds to a domain and contains clients belonging to this domain. Next we aggregate the clients within each $\beta_u^t$ to identify domain experts $w_u^{t,dom}$. The process can be formulated as:

$$\boldsymbol{v} = [\boldsymbol{v}_1, \boldsymbol{v}_2, \boldsymbol{v}_3, \boldsymbol{v}_4, \boldsymbol{v}_5, \boldsymbol{v}_6]$$
$$\Downarrow \ Cluster$$
$$= [\underbrace{\boldsymbol{v}_1, \boldsymbol{v}_2}_{\text{Domain 1}}, \underbrace{\boldsymbol{v}_3, \boldsymbol{v}_4}_{\text{Domain 2}}, \underbrace{\boldsymbol{v}_5, \boldsymbol{v}_6}_{\text{Domain 3}}] \tag{13a}$$
$$\Downarrow \ Map \ to \ client \ models$$
$$\boldsymbol{\omega} = [\underbrace{\boldsymbol{\omega}_1, \boldsymbol{\omega}_2}_{\beta_1}, \underbrace{\boldsymbol{\omega}_3, \boldsymbol{\omega}_4}_{\beta_2}, \underbrace{\boldsymbol{\omega}_5, \boldsymbol{\omega}_6}_{\beta_3}],$$

$$w_u^{t,dom} = \frac{\sum\limits_{w_k^{t-1} \in \beta_u} \alpha_k w_k^{t-1}}{\sum\limits_{w_k^{t-1} \in \beta_u} \alpha_k}. \tag{13b}$$

Thus clients can get rich domain knowledge from these domain experts. We calculate the logits output through $k^{th}$ client model and each domain expert $w_u^{t,dom}$ on private data $x_i$ w.r.t its ground truth label $y_i$: $Z_{i,k}^t = f(w_k^t, x_i)$ and $Z_{i,u}^{t,dom} = f(w_u^{t,dom}, x_i)$. By inserting $Z_{i,k}^t$ and $Z_{i,u}^{t,dom}$ into the Eq.(11), We get the final defined knowledge distillation loss function $\mathcal{L}_{fkd}$:

$$\mathcal{L}_{fkd} = \sum_u \frac{1}{|\beta^t|}[\mathcal{L}_{dkd}(Z_{i,k}^t, Z_{i,u}^{t,dom})] \tag{14}$$

Eq.(14) assigns each expert the same weight thus mitigating the issue of domain skew. This dynamic knowledge distillation method reliably and efficiently distills knowledge for each domain. Combined with the cross-entropy loss $\mathcal{L}_{CE}$, the local loss of client $k$ is now defined as:

$$\mathcal{L}_k = \mathbb{E}_{(x_i,y_i)\sim\mathcal{D}_k}(\mathcal{L}_{CE} + c\mathcal{L}_{fkd}), \tag{15}$$

where c represents the knowledge distillation intensity of the method.

### 3.4  Discussion and Limitation

**Clustering Technical.** A variety of clustering techniques have been proposed to discover natural grouping [55, 7, 8, 50, 33, 45]. The well-known methods, K-Means [37] and DBSCAN iteratively assign points to a fixed group number. However, they are sensitive to hyper-parameter selection under different scenarios. Thus we shift the gaze towards FINCH [45], which is parameter-free and thus

Table 1: **Ablation** on popular clustering methods. Please refer to Sec.3.4 for details.

| Methods | Office31 | | | |
|---|---|---|---|---|
| | A | W | D | AVG |
| K-means | 66.89 | 44.31 | 28.24 | 46.48 |
| DBSCAN | 65.28 | 36.82 | 28.76 | 43.62 |
| **FINCH (ours)** | **67.06** | **54.82** | **29.84** | **50.57** |

suitable for heterogeneous federated learning. Specifically, we leverage the cosine similarity metric to evaluate the distance between any two client weights and view the weight with minimum distance as its "neighbor", sorted into the same set. After clustering, we aggregate all clients in the same cluster as they have related domain knowledge in order to get domain experts. We compare FINCH [45] with the well-known clustering methods, K-Means [37] and DBSCAN [8]. The results are shown in Tab.1.

**Conceptual Difference.** Unlike conventional federated KD methods that depend solely on forward KLD and a single teacher model, DKDR introduces two distinctive innovations: dynamic KLD weighting and the use of multiple domain experts. The dynamic weighting reduces distillation bias by adapting to knowledge discrepancies, which is a departure from the static approaches of methods like FedNTD [25] or FedX [11]. It ensures precise alignment with the target distribution across diverse domains. Furthermore, by identifying domain experts, DKDR provides tailored guidance to clients, overcoming the limitation that the single teacher model may underperform in specific domains. This multi-expert paradigm enhances reliability and performance in multi-domain FL scenarios.

**Limitation.** While DKDR effectively enhances the reliability of distillation pathways and teacher models in multi-domain federated learning, it is not without drawbacks. The dynamic weighting mechanism involves SVD and clustering techniques, adding computational overhead that may significantly prolong training, particularly on resource-limited clients. Additionally, DKDR assumes that domains are sufficiently distinct for clustering to accurately identify experts; if domains overlap significantly, this assumption falters, potentially degrading overall performance.

## 4  Experiments

### 4.1  Experimental Setup

**Datasets.** We evaluate DKDR on two single-domain scenarios and two multi-domain scenarios.

- **Cifar-10** [23] contains 50k training images and 10k test images with $32\times32$ for 10 classes.
- **Cifar-100** [23] contains $50k$ and $10k$ images with $32\times32$ for 100 classes.
- **Office31** [44] consists of three domains: Amazon (A), Webcam (W) and DSLR (D). In total, the dataset contains 4,110 images with $256\times256$ across 31 categories, shared among the three domains.
- **Office Home** [51] consists of four domains: Art (A), Clipart (C), Product (P), and Real World (R). It contains 15,500 images with $256\times256$ across 65 categories, shared among the four domains.

**Data Heterogeneity**. As for the data heterogeneity simulation, we utilize the Dirichlet distribution, $Dir(\zeta)$ to simulate the label skew, as previous methods [29, 28, 63]. The smaller $\zeta$ is, the more imbalanced the local distribution is.

**Counterparts.** We compare our method with state-of-the-art (SOTA) federated knowledge distillation and federated learning approaches: FedAvg [39], FedProx [29], FedDyn [2], Scaffold [19], FedProto [49],MOON [28], FedNTD [25],FedDf [32].

**Implement Details.** We conduct communication epoch for $E = 200$ and local updating round $T = 5$, where all federated learning approaches have little or no accuracy gain with more communications. We use the SGD optimizer with the learning rate $lr = 1e - 3$. The corresponding weight decay is $1e - 5$ and momentum is $0.9$. The training batch size is $64$ for single-domain tasks and $16$ for multi-domain tasks. The client number $K$ is 20 for different datasets. We conduct experiments with ResNet-10 [12] on single-domain scenarios and ResNet-18 [12] on multi-domain scenarios. We fix the random seed to ensure reproduction and conduct experiments on the NVIDIA 3090Ti.

Table 2: **Comparison with the state-of-the-art method in the Office31 and Office Home** with domain skew. Best in bold and second with underline. Please refer to Sec.4.2 for further explanations.

| Methods | Office31 | | | | Office Home | | | | |
|---|---|---|---|---|---|---|---|---|---|
| | A | W | D | AVG | A | C | P | R | AVG |
| FedAvg | 48.04 | 32.91 | 22.45 | 34.47 | 39.24 | 61.29 | 74.58 | 58.45 | 58.39 |
| FedProx | 45.55↓2.49 | 26.58↓6.33 | 19.39↓3.06 | 30.51↓3.96 | 39.67↑0.43 | 61.12↓0.17 | 74.52↓0.06 | 59.66↑1.21 | 58.74↑0.35 |
| FedDyn | 56.58↑8.54 | 20.25↓12.66 | 23.47↑1.02 | 33.43↓1.04 | 39.26↑0.02 | 60.21↓1.08 | 73.84↓0.74 | 59.08↑0.63 | 58.10↓0.29 |
| Scaffold | 46.80↓1.24 | 34.18↑1.27 | 23.47↑1.02 | 34.82↑0.35 | 39.88↑0.64 | 61.24↓0.05 | 74.75↑0.17 | 58.85↑0.60 | 58.68↑0.29 |
| FedProto | 51.60↑3.56 | 36.71↑3.80 | 31.63↑9.18 | 39.98↑5.51 | 40.29↑1.05 | 63.65↑2.36 | 75.31↑0.73 | 60.34↑1.89 | 59.90↑1.51 |
| MOON | 49.47↑1.43 | 32.28↓0.63 | 26.53↑4.08 | 36.09↑1.62 | 38.78↓0.46 | 62.09↑0.80 | 74.27↓0.31 | 58.90↑0.45 | 58.51↑0.12 |
| FedNTD | 48.22↑0.18 | 35.44↑2.53 | 22.45↓0.00 | 35.37↑0.90 | 39.28↑0.04 | 63.07↑1.78 | 75.20↑0.62 | 59.20↑0.75 | 59.18↑0.79 |
| FedDf | 45.26↓2.78 | 32.78↓0.13 | 24.72↑2.27 | 34.25↓0.22 | 37.93↓1.31 | 60.53↓0.76 | 71.94↓2.64 | 57.83↑0.62 | 57.06↓1.33 |
| DKDR | **67.06**↑19.02 | **54.82**↑21.91 | 29.84↑7.39 | **50.57**↑16.10 | **42.68**↑3.44 | **65.99**↑4.70 | **78.22**↑3.64 | **63.03**↑4.58 | **62.48**↑4.09 |

## 4.2 Comparison to State-of-the-Arts

**Performance Comparison** The Tab.2 , Tab.3 and Tab.4 present the final accuracy metric by the end of the federated learning process with popular SOTA methods. It depicts that our method outperforms all other baselines in seven out of the eight settings, which confirms that the knowledge distillation of DKDR is reliable, efficient, and possesses superior domain generalization capabilities.

Table 3: **Comparison with the state-of-the-art method in the Cifar-10** with skew ratio $\zeta \in \{0.1, 0.3, 0.5\}$. Please refer to Sec.4.2 for details.

| Methods | Cifar-10 | | |
|---|---|---|---|
| | $\zeta = 0.1$ | $\zeta = 0.3$ | $\zeta = 0.5$ |
| FedAvg | 76.91 | 79.86 | 80.34 |
| FedProx | 70.43↓6.48 | 74.14↓5.72 | 75.11↓5.23 |
| FedDyn | 78.77↑1.86 | 80.79↑0.93 | 81.26↑0.92 |
| Scaffold | **79.62**↑2.71 | 80.99↑1.13 | 81.40↑1.06 |
| FedProto | 78.45↑1.54 | 80.13↑0.27 | 81.49↑1.15 |
| MOON | 75.24↓1.67 | 79.83↓0.03 | 80.71↑0.37 |
| FedNTD | 77.18↑0.27 | 80.36↑0.50 | 80.94↑0.60 |
| FedDf | 78.53↑1.62 | 80.24↑0.38 | 81.37↑1.03 |
| DKDR | 79.46↑2.55 | **81.93**↑2.07 | **82.63**↑2.29 |

Table 4: **Comparison with the state-of-the-art method in the Cifar-100** with skew ratio $\zeta \in \{0.1, 0.3, 0.5\}$. Please refer to Sec.4.2 for details.

| Methods | Cifar-100 | | |
|---|---|---|---|
| | $\zeta = 0.1$ | $\zeta = 0.3$ | $\zeta = 0.5$ |
| FedAvg | 43.76 | 46.66 | 48.57 |
| FedProx | 33.92↓9.84 | 37.81↓8.85 | 39.79↓8.78 |
| FedDyn | 46.21↑2.45 | 48.82↑2.16 | 50.23↑1.66 |
| Scaffold | 46.32↑2.56 | 50.33↑3.67 | 51.76↑3.19 |
| FedProto | 44.21↑0.45 | 49.88↑3.22 | 51.34↑2.77 |
| MOON | 42.96↓0.80 | 45.73↓0.93 | 48.58↑0.01 |
| FedNTD | 44.12↑0.36 | 47.10↑0.44 | 48.99↑0.42 |
| FedDf | 45.12↑1.36 | 46.62↓0.04 | 48.97↑0.40 |
| DKDR | **46.80**↑3.04 | **51.84**↑5.18 | **53.18**↑4.61 |

**Convergence Analysis** Fig.4 shows the curves of the average test accuracy during the training process across three random runs of three datasets (Cifar-100, Office31, Office Home) representing single-domain and multi-domain scenarios, including the results of various baselines. Traditional FL methods such as FedAvg performs poorly in heterogeneous scenarios while methods designed specifically for heterogeneous problem such as Scaffold and FedProto achieve much better performance.

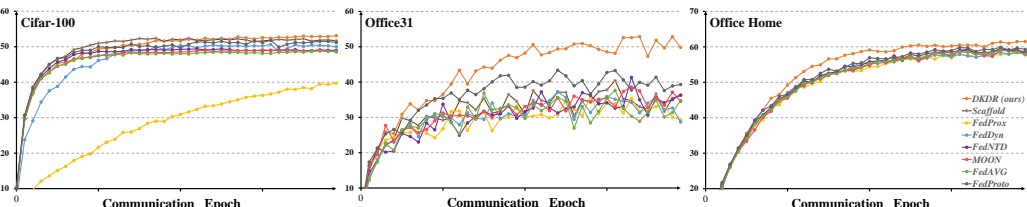

Figure 4: **Visualization** of training curves of the average test accuracy of DKDR and various baselines on three datasets (Cifar-100, Office31, Office Home). Please refer to Sec.4.2 for further explanations.

## 4.3 Sensitivity

**Hyper Parameters $c$ and $\mu$.** In the single-domain task Cifar-100 and the multi-domain task Office31, the optimal values of $c$ and $\mu$ remain stable at 1.25 and 0.5 across different scenarios. This aligns with our theory (Sec.3.2): when $\mu < 0.5$, DKD gradually degenerates into forward KLD, while when $\mu > 0.5$, it gradually degenerates into reverse KLD. Both cases introduce different biases.

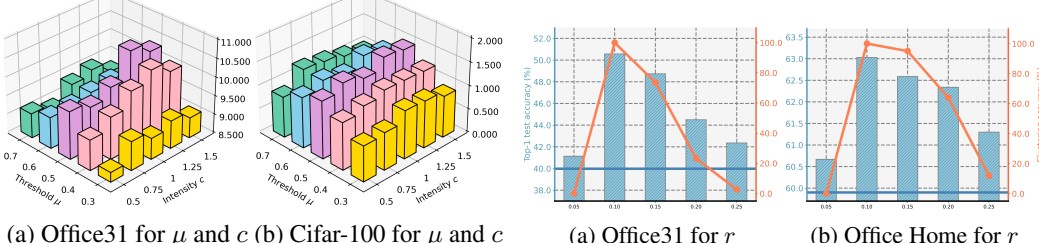

(a) Office31 for $\mu$ and $c$ (b) Cifar-100 for $\mu$ and $c$      (a) Office31 for $r$     (b) Office Home for $r$

Figure 5: **Sensitivity analysis for $\mu$ and $c$ on Office31 and Cifar-100.** The z-axis represents the performance improvement relative to the baselines. Please refer to Sec.4.3 for further detailed explanations.

Figure 6: **Sensitivity analysis for $r$** across two multi-domain tasks by Top-1 test Accuracy and Clustering accuracy. The horizontal line represents the baseline. Please refer to Sec.4.3 for further explanations.

**Hyper Parameter $r$.** In assessing the filtering strength $r$ of SVD, we utilize the average accuracy of clustering within each domain additionally. If a cluster includes clients from more than one domain, it is considered a clustering failure for all clients in that cluster. As shown in Fig.6a and Fig.6b, the optimal setting of $r$ remains stable at 0.1 for both two datasets. It is worth noting that when too few singular values are retained, the domain signals tend to converge towards a common low-dimensional subspace, which can lead to the failure of clustering, just as when $r = 0.05$.

## 4.4 Effectiveness.

**Effects of Key Components Mechanism of DKDR.** To substantiate its robustness and stability, we meticulously evaluate the performance across both single-domain and multi-domain scenarios. As illustrated in Tab.5, compared to FedAVG, DKD achieves a consistent performance enhancement in both tasks, whereas KDP demonstrates more pronounced effectiveness in complex multi-domain tasks, as domain-specific experts can provide reliable guidance for clients in each domain.

**Ablation Study of DKD.** We compare DKD with KD using forward and reverse KLD separately to validate the effectiveness of DKD. As shown in Tab.6, DKD is more suitable for federated learning tasks in both single-domain and multi-domain tasks compared to using forward or reverse KLD alone.

Table 5: Ablation study of the key components in DKDR on four datasets (Office31, Office Home, Cifar-10, Cifar-100). Please see Sec.4.4 for details.

Table 6: Ablation study of the DKD of four datasets (Office31, Office Home, Cifar-10, Cifar-100). Please refer to Sec.4.4 for further detailed explanations.

| DKD | KDP | Office31 | Office Home | Cifar-10 | Cifar-100 |
|---|---|---|---|---|---|
| ✗ | ✗ | 34.47 | 58.39 | 79.86 | 46.66 |
| ✓ | ✗ | 36.83 | 59.52 | 80.72 | 48.79 |
| ✗ | ✓ | 48.59 | 61.03 | 80.63 | 50.33 |
| ✓ | ✓ | **50.57** | **62.48** | **81.23** | **51.84** |

| FKD | RKD | DKD | Office31 | Office Home | Cifar-10 | Cifar-100 |
|---|---|---|---|---|---|---|
| ✗ | ✗ | ✗ | 34.47 | 58.39 | 79.85 | 46.66 |
| ✓ | ✗ | ✗ | 34.88 | 58.70 | 80.59 | 47.52 |
| ✗ | ✓ | ✗ | 35.67 | 58.61 | 80.13 | 48.03 |
| ✗ | ✗ | ✓ | **36.83** | **59.52** | **80.72** | **48.79** |

## 5 Conclusion

In this paper, we address two significant problems in existing federated knowledge distillation methods: the unreliability of distilling pathway and teacher model. We empirically and theoretically analyze the fundamental differences between forward and reverse KLD, which leads us to propose a dynamic distillation approach that minimize distillation bias. To get reliable guidance, we employed knowledge decoupling to identify domain experts. Based on these insights, we propose the DKDR framework, which is strategically designed to achieve robust performance across diverse tasks. The effectiveness of DKDR has been validated with many sota methods over various classification tasks.

## Acknowledgement

This work is supported by National Natural Science Foundation of China under Grant (62361166629, 623B2080, 62506269), the Major Project of Science and Technology Innovation of Hubei Province (2024BCA003, 2025BEA002), and the Innovative Research Group Project of Hubei Province under Grants 2024AFA017. The supercomputing system at the Supercomputing Center of Wuhan University supported the numerical calculations in this paper. This research was financially supported by the Open Research Fund from Guangdong Laboratory of Artificial Intelligence and Digital Economy (SZ), under Grant No.GML-KF-24-10.

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

# A  Convergence Proof of DKDR

## A.1  Assumptions

L-smoothness: For all $k$, $f_k(w)$ is $L$-smooth: $\|\nabla f_k(w_1) - \nabla f_k(w_2)\| \leq L\|w_1 - w_2\|$.

(Dynamic weights preserve smoothness; $\gamma_s, \gamma_l$ changes Lipschitz: $|\gamma_s(w_1) - \gamma_s(w_2)| \leq \kappa\|w_1 - w_2\|$, yielding $L' \leq L + \kappa \max(G_{skd}, G_{rkd})$. Quasi-static: $\gamma$ per round.)

Bounded gradients: $\|\nabla f_k(w)\|^2 \leq G^2$.

Bounded variance: $\mathbb{E}_k[\|\nabla f_k(w) - \nabla f(w)\|^2] \leq \sigma^2$.

Bounded distillation gradients: $\|\nabla L_k^{DKD}(w)\|^2 \leq D^2 \leq \max(G_{skd}^2, G_{rkd}^2)$ (logits clipped).

Non-convex: $f(w)$ lower-bounded by $f^*$.

Parameters: $\eta > 0$, rounds $T$, local steps $E$, $\eta \leq 1/(4L'E)$.

## A.2  Theorem 1 (Convergence of DKD)

$$\frac{1}{T}\sum_{t=0}^{T-1}\mathbb{E}\|\nabla f(w^t)\|^2 \leq \frac{4(f(w^0) - f^*)}{\eta TE} + 6L'\eta(E-1)G^2 + \frac{\sigma^2}{K} + 8(L')^2 E\eta\left(G^2 + \frac{\Lambda(\gamma)}{K}\right) + O(\kappa^2\eta^2 EG^2),$$

where $\Lambda(\gamma) = \mathbb{E}_k[\|\nabla f_k(w) - \nabla f(w)\|^2]$.

Dynamic weights yield $\Lambda(\gamma) < \Lambda_0$ (static). For $\eta = O(\sqrt{K/(TEL')})$, RHS $O(1/\sqrt{KTE})$, converges to 0 as $T \to \infty$. Compared to FedAvg ($\Lambda = \sigma^2$) or static KLD, DKD tighter.

**Key Lemma.** Reduction in $\Lambda(\gamma)$ Forward KLD prioritizes $\gamma_l$ (mean-seeking; $\partial L_{skd}/\partial z_{j,k} = z_{j,k} - z_j^{t-1}$); reverse prioritizes $\gamma_s$ (mode-seeking; $\partial L_{rkd}/\partial z_{j,k} = z_{j,k}\log(z_{j,k}/z_j^{t-1}) - L_{rkd}$). Weights $\alpha = \gamma_l/(\gamma_s + \gamma_l)$ balance via gradient norm.

$$\nabla L_{dkd} = \frac{\gamma_s}{\gamma_s + \gamma_l}\nabla L_{rkd} + \frac{\gamma_l}{\gamma_s + \gamma_l}\nabla L_{skd}.$$

By Pinsker's ($KL \geq (1/2)\,TV^2$):

$$\mathbb{E}[KL(Z_k\|Z_{global})] \approx \min_{path}\int \|\nabla L_{dkd}\|dt \leq \max(\mathbb{E}[KL_{skd}], \mathbb{E}[KL_{rkd}]) - \delta\frac{\gamma_s\gamma_l}{(\gamma_s + \gamma_l)^2},$$

$\delta > 0$ from complementarity. Specifically,

$$\|\nabla L_{dkd}\|^2 \leq \max(\|\nabla L_{skd}\|^2, \|\nabla L_{rkd}\|^2) - \beta\|\nabla L_{skd} - \nabla L_{rkd}\|^2\frac{\gamma_s\gamma_l}{(\gamma_s + \gamma_l)^2},$$

($\beta = \Theta(1)$). Thus,

$$\|\nabla f_k(w) - \nabla f(w)\|^2 \leq \Lambda_0 - \beta\frac{\gamma_s\gamma_l}{(\gamma_s + \gamma_l)^2},$$

$$\Lambda(\gamma) \leq \Lambda_0(1 - \epsilon)(\epsilon \approx 1/2 \text{ when balanced}) < \Lambda_0.$$

## A.3  Proof Sketch

**Step 1**: Local:
$$w_k^{t,e+1} = w_k^{t,e} - \eta g_k^{t,e}, g_k^{t,e} = \nabla f_k(w_k^{t,e}).$$

Aggregate:
$$w^{t+1} = (1/K)\sum w_k^{t,E}.$$

Define
$$\bar{w}^{t,e} = (1/K)\sum w_k^{t,e}, \bar{g}^{t,e} = (1/K)\sum g_k^{t,e}.$$

**Step 2**: Descent:

$$\mathbb{E}[f(w^{t+1})] \leq f(w^t) + \langle \nabla f(w^t), w^{t+1} - w^t \rangle + (L'/2)\|w^{t+1} - w^t\|^2,$$

$$w^{t+1} - w^t = -\eta \sum_{e=0}^{E-1} \bar{g}^{t,e}/E,$$

yielding:

$$\leq f(w^t) - (\eta E/2)\|\nabla f(w^t)\|^2 + (\eta/(2E)) \sum_{e=0}^{E-1} \mathbb{E}\|\bar{g}^{t,e} - \nabla f(w^t)\|^2 + (L'\eta^2 EG^2/2)$$

.

**Step 3**: Bias:

$$\mathbb{E}\|\bar{g}^{t,e} - \nabla f(w^t)\|^2 \leq (\sigma^2 + \Lambda(\gamma))/K + 6(L')^2\eta^2 e(G^2 + \Lambda(\gamma)/K) + O(\kappa^2\eta^2 eG^2)$$

.

**Step 4**: Sum t=0 to T-1, divide by $\eta ET/2$; $\sum e \leq E^2/2$, constants to 8, yields bound.

## B  Notations Table

| Symbol | Meaning | Symbol | Meaning |
|--------|---------|--------|---------|
| $Z_w(y \mid x)$ | Local model distribution | $Z(y \mid x)$ | Global model distribution |
| $D_k$ | Private data of $k$-th client | $N_k$ | Number of data points |
| $w^t$ | Global model parameters at round $t$ | $w_k^t$ | Local model parameters of $k$-th client |
| $Z_{i,k}^t$ | Client model output distribution | $Z_i^{t-1}$ | Global model output distribution |
| $\sigma$ | Softmax function | $\mathcal{L}_{\text{skd}}$ | Standard KD loss |
| $\mathcal{L}_{rkd}$ | Reverse KD loss | $a(j)$ | Indicator function for DKCs/AKCs |
| $\mu$ | Threshold to define DKCs/AKCs | $\gamma_s$ | Knowledge discrepancy in AKCs |
| $\gamma_l$ | Knowledge discrepancy in DKCs | $\mathcal{L}_{dkd}$ | Dynamic KD loss |
| $v_k^t$ | Difference vector | $\mathbf{U}, \mathbf{\Sigma}, \mathbf{V}^T$ | SVD decomposition matrices |
| $\beta^t$ | Client clusters | $w_v^{t,\text{dom}}$ | Domain expert model |
| $\mathcal{L}_{fkd}$ | Final KD loss | $\mathcal{L}_k$ | Local loss of $k$-th client |

