# OpenReview forum: "DKDR: Dynamic Knowledge Distillation for Reliability in Federated Learning"
_NeurIPS.cc/2025/Conference — NeurIPS 2025 poster_

### Official Review · Reviewer_YmgD · 2025-06-16

**Clarity:** 3
**Significance:** 3
**Originality:** 3
**Rating:** 5
**Confidence:** 5

**Summary:**

To address the issue of unreliability in knowledge distillation within federated learning, this paper proposes DKDR, a federated learning framework with two key innovations: dynamically weighting forward/reverse KLD and knowledge decoupling for domain-experts identification. Extensive experiments across various federated settings demonstrate DKDR’s superior performance over SOTA methods.

**Questions:**

The authors mention that prevalent distillation solutions primarily minimize forward KLD to fit global model. Can you elaborate on why it results in significant distillation bias?

**Ethical Concerns:**

["NO or VERY MINOR ethics concerns only"]

**Final Justification:**

Thank you for your comprehensive and thoughtful response to my concerns. I appreciate the effort you’ve put into addressing each point with detailed explanations, quantitative analyses, and additional experimental results.

Your clarification on computational overhead effectively addresses my initial worries about practical feasibility. The quantitative evidence showing that computational and communication costs remain within acceptable ranges, even with varying client numbers, is particularly convincing.

I also find your analysis of domain overlap scenarios compelling. The explanation of how hierarchical clustering and dynamic distillation mechanisms enable graceful degradation, combined with ablation results demonstrating the resilience of the DKD module, helps alleviate concerns about the method’s robustness in real-world heterogeneous settings.

Finally, your theoretical and empirical explanation of why forward KLD introduces distillation bias in multi-domain scenarios clarifies a key motivation for your work, enhancing the depth of your contribution.

Based on these thorough responses and the additional evidence provided, I am pleased to increase my score. I encourage you to incorporate these clarifications and experimental details into the final version of the paper to strengthen its overall impact.

**Limitations:**

See weakness.

**Paper Formatting Concerns:**

None.

**Quality:**

3

**Strengths And Weaknesses:**

**Strengths:**
- The proposed method of federated KD appears interesting and innovative.
- The paper presents a clear and concise writing style thus easy to follow.
- The dynamic weighting of forward and reverse KLD is a novel approach to mitigate distillation bias with both theoretical and empirical validation.
- The analysis of forward vs. reverse KLD behaviors provides a clear foundation for the method’s design.

**Weaknesses:**
- SVD and clustering add computational overhead, potentially limiting scalability on resource-constrained devices.
- The method assumes distinct domains, which may falter with significant overlap.
- The work lacks analysis of model performance with different local training epochs for each communication round.

---

> ### Author Rebuttal · Authors · 2025-07-30
>
> ## Response to Reviewer YmgD
>
> Dear Reviewer YmgD,
>
> Thank you very much for your thoughtful and thorough review. We are encouraged by your recognition of our innovative approach to federated knowledge distillation. We appreciate your careful examination of our work and hope our detailed responses below will address your concerns.
>
> ---
>
> ### Weaknesses
>
> >**`W1: Computational Overhead Discussion.`**
>
> **A1:** Thank you for this important practical concern. We want to clarify the computational distribution and provide quantitative analysis:
>
> **Computational Distribution:**
> - **Client-side overhead**:
>     - Only involves simple arithmetic for dynamic weight calculation: $\frac{\gamma\_s}{\gamma\_s+\gamma\_l}$ and $\frac{\gamma\_l}{\gamma\_s+\gamma\_l}$
>     - No SVD or clustering operations on client devices
> - **Server-side operations**:
> 	- SVD and FINCH clustering perform once per communication round
>     - Typical federated learning servers have sufficient computational resources
>
> **Further Analysis:**
> We conduct experiments in OfficeCaltech to further analyze it, as shown in the below table.
>
> *Table. **Comparison of computational and communication costs** in OfficeCaltech dataset.*
>
> | Method   | Client Time (s) | Server Time (s) | Communication Cost (mb) | Accuracy (%) |
> | -------- | --------------- | --------------- | ----------------------- | ------------ |
> | FedAVG   | 2.70            | 0.02            | 2.83                    | 77.94        |
> | FedProto | 14.13           | 0.01            | 2.83                    | 79.26        |
> | Scaffold | 9.23            | 0.02            | 8.48                    | 75.32        |
> | **DKDR** | 6.28            | 0.11            | 8.25                    | **81.14**    |
>
> We also conduct experiments on varying client numbers, as shown in the following table. The computational and communication overheads are within an acceptable range, and the performance improvements are worthwhile.
>
> *Table. **The relationship of client Number $K$ and computation cost** in OfficeCaltech dataset.*
>
> | Client Number $K$ | Server Processing Time (s) | Improvement (Compared to FedAVG) |
> | ----------------- | -------------------------- | -------------------------------- |
> | 20                | 0.11                       | +3.20%                           |
> | 60                | 0.42                       | +2.06%                           |
> | 100               | 2.94                       | +2.54%                           |
>
>
>
> >**`W2: Discussion on Domain Overlap Scenarios.`**
>
> **A2:** Thank you for highlighting this important assumption. We acknowledge that significant domain overlap poses challenges to our knowledge decoupling approach. However, we have conduct analysis showing that DKDR exhibits **graceful degradation** rather than catastrophic failure in such scenarios:
>
> **Robustness Under Domain Overlap:**
> When domain boundaries become unclear due to significant overlap, our method handles this situation through multiple mechanisms:
> 1. **Hierarchical Clustering Resilience**: FINCH clustering naturally accommodates overlapping structures by creating hierarchical representations, allowing it to identify meaningful groupings even when domains are not perfectly distinct.
> 2. **Fallback to Dynamic Distillation**: Even when knowledge decoupling produces suboptimal domain experts due to overlap, the **Dynamic Knowledge Distillation (DKD) component continues to provide substantial benefits**. The dynamic weighting mechanism adapts to knowledge discrepancies regardless of domain expert quality.
> 3. **Graceful Performance Degradation**: In worst-case scenarios where clustering completely fails to separate domains, DKDR essentially reduces to an enhanced federated distillation method that still outperforms standard approaches due to the DKD component.
>
> **Empirical Validation:**
> Our ablation studies (Sec.4.4, Page 9) demonstrate that even when the KDP module is disabled (✗ KDP), the DKD module alone (✓ DKD) still provides consistent improvements across all datasets, confirming the method's resilience to domain overlap issues.
>
>
>
> >**`W3: Abalation Study of Local Epoch.`**
>
> **A3:** Excellent point! We conduct additional experiments analyzing the sensitivity of DKDR to local training epochs:
>
> *Table. **Ablation study of Local Epochs T** on Digits dataset.*  DKDR achieves consistent better performance compared to three baselines (FedAVG, FedProto, Scaffold) in 4 of 5 settings.
>
> | Local Epochs (T) | FedAvg | FedProto | Scaffold  | DKDR      |
> | ---------------- | ------ | -------- | --------- | --------- |
> | T=1              | 63.99  | 63.56    | 53.43     | **64.53** |
> | T=3              | 64.32  | 63.24    | 61.13     | **65.36** |
> | T=5              | 62.85  | 62.84    | **67.04** | 66.83     |
> | T=10             | 60.13  | 61.90    | 65.20      | **67.24** |
> | T=20             | 58.56  | 60.90    | 65.62     | **65.73** |
>
>
>
> ---
> ### Questions
>
> >**`Q1: Why minimizing forward KLD results in significant distillation bias?`**
>
> **A1:** Excellent question that goes to the heart of our motivation! The significant distillation bias from forward KLD arises from its **mean-seeking behavior** in multi-domain federated scenarios. Let us elaborate with both theoretical and empirical perspectives:
>
> **Theoretical Foundation:**
> Forward KLD aims to minimize: $KL[Z^{t-1}||Z^t\_{k}] = \sum\_j z^{t-1}\_j \log\frac{z^{t-1}\_j}{z^t\_{j,k}}$
> From the gradient analysis (Eq. 8a): $\frac{\partial L\_{fkd}}{\partial z^t\_{j,k}} = z^t\_{j,k} - z^{t-1}\_j$
>
> This shows that **larger $z^{t-1}\_j$ values receive proportionally higher weights** in the loss function. When the global model produces multi-peaked distributions (common in heterogeneous FL), the mean-seeking behaviour of forward KLD forces the local model $z^t\_{j,k}$ to cover all peaks of the global distribution $z^{t-1}\_j$, including low-density regions between peaks.
>
> **Concrete Example:**
> Consider a scenario where:
> - Client A specializes in cats (peak at class 3)
> - Client B specializes in dogs (peak at class 7)
> - Global model after aggregation shows peaks at both classes 3 and 7
>
> When Client A uses forward KLD to distill from this global model, forward KLD forces Client A to assign probability to both cats and dogs. This creates **unreasonable high probabilities in low-density regions** (the valley between peaks), which results in bias.
>
> **Empirical Evidence:**
> Our experiments (Figure 2) demonstrate this phenomenon. Forward KLD shows higher knowledge discrepancies in AKCs (lower-probability regions) while reverse KLD shows higher discrepancies in DKCs (dominant regions), which validates that forward KLD struggles with multi-peaked distributions.

---

> ### Comment · Reviewer_YmgD · 2025-08-05
>
> Thank you for your comprehensive and thoughtful response to my concerns. I appreciate the effort you’ve put into addressing each point with detailed explanations, quantitative analyses, and additional experimental results.
>
> Your clarification on computational overhead effectively addresses my initial worries about practical feasibility. The quantitative evidence showing that computational and communication costs remain within acceptable ranges, even with varying client numbers, is particularly convincing.
>
> I also find your analysis of domain overlap scenarios compelling. The explanation of how hierarchical clustering and dynamic distillation mechanisms enable graceful degradation, combined with ablation results demonstrating the resilience of the DKD module, helps alleviate concerns about the method’s robustness in real-world heterogeneous settings.
>
> Finally, your theoretical and empirical explanation of why forward KLD introduces distillation bias in multi-domain scenarios clarifies a key motivation for your work, enhancing the depth of your contribution.
>
> Based on these thorough responses and the additional evidence provided, I am pleased to increase my score. I encourage you to incorporate these clarifications and experimental details into the final version of the paper to strengthen its overall impact.

---

> > ### Author Response · Authors · 2025-08-06
> >
> > Dear Reviewer YmgD,
> >
> > Thank you for your comprehensive review and constructive comments on our submission. Your feedback has been invaluable in enhancing the clarity and impact of our work. We greatly appreciate the opportunity to address your suggestions and your dedication to reviewing our manuscript.
> >
> > Best regards,
> >
> > Authors

---

### Official Review · Reviewer_rkk9 · 2025-06-25

**Clarity:** 3
**Significance:** 3
**Originality:** 4
**Rating:** 6
**Confidence:** 5

**Summary:**

The paper presents DKDR. It dynamically weights forward and reverse KLD for reliable distillation and decouples knowledge to get reliable teachers. Experiments on image classification tasks show its superiority.

**Questions:**

Q1.	Extreme Heterogeneity: How does DKDR perform under more severe data heterogeneity? Could the authors provide additional experiments to explore this?
Q2.	Scalability: How does the method scale with a larger number of clients? Are there optimizations to handle such scenarios efficiently?

**Ethical Concerns:**

["NO or VERY MINOR ethics concerns only"]

**Final Justification:**

The authors has addressed most of my previous concerns. By considering with other reviewers' comments, I decide to raise my score to 6.

**Limitations:**

See weaknesses above.

**Paper Formatting Concerns:**

There are no issues regarding paper formatting.

**Quality:**

3

**Strengths And Weaknesses:**

Strengths:
S1.	Innovative Approach: The dynamic weighting of forward and reverse KLD based on knowledge discrepancies is a creative and effective solution to mitigate distillation bias in multi-domain FL, as supported by theoretical and empirical analysis.
S2.	Easy to follow: The paper is well-written and easy to follow. Moreover, the precise description of modules with detailed exquisite framework visualization makes the overall work clear.
S3.	Well-motivated: The proposed method appears well-motivated and effectively addresses the identified challenges in federated learning.
Weaknesses:
W1.	Computational Overhead: The use of SVD and clustering introduces additional computational complexity, which may be burdensome for resource-constrained clients, though this is acknowledged in the limitations.
W2.	Limited Heterogeneity Extremes: The paper does not explore performance under extreme data heterogeneity, which could further demonstrate robustness.
W3.	Scalability Concerns: Experiments are limited to 20 clients; scalability with hundreds or thousands of clients, common in real-world FL, is not addressed.

---

> ### Author Rebuttal · Authors · 2025-07-30
>
> ## Response to Reviewer rkk9
>
> Dear Reviewer rkk9,
>
> Thank you very much for your positive and constructive review. Your acknowledgment of our technical soundness and high impact are deeply appreciated. We hope our responses below will further strengthen your confidence in our work.
>
> ---
>
> ### Weaknesses
>
> >**`W1: Computational Overhead of DKDR.`**
>
> **A1:** Sorry for the misunderstanding. We want to clarify that the **computational overhead occurs primarily on the server side, not on resource-constrained clients**. We will fix it in the final version. Here is the detailed breakdown:
>
> **Client-side Operations**: Clients only perform standard local training and knowledge distillation using the dynamic weighting mechanism. The dynamic weight calculation ($γ\_s/(γ\_s+γ\_l)$ and $γ\_l/(γ\_s+γ\_l)$) involves simple arithmetic operations with negligible overhead.
>
> **Server-side Operations**: SVD decomposition and FINCH clustering are performed once per communication round on the server, which typically has sufficient computational resources in federated learning scenarios.
>
> **Quantitative Analysis**: We conduct overhead analysis on the OfficeCaltech dataset, as shown in the following table.
>
> *Table. **Comparison of Time and Memory Usage** in OfficeCaltech dataset.*
>
> | Method   | Client Time (s) | Server Time (s) | Communication Cost (mb) | Accuracy (%) |
> | -------- | --------------- | --------------- | ----------------------- | ------------ |
> | FedAVG   | 2.70            | 0.02            | 2.83                    | 77.94        |
> | FedProto | 14.13           | 0.01            | 2.83                    | 79.26        |
> | Scaffold | 9.23            | 0.02            | 8.48                    | 75.32        |
> | **DKDR** | 6.28            | 0.11            | 8.25                    | **81.14**    |
>
>
>
>
> >**`W2 & Q1: Data Heterogeneity Ablation.`**
>
> **A2:** Excellent point! We conduct additional experiments under extreme heterogeneity settings. We use Dirichlet parameter $ζ=0.05$ (much smaller than our original $ζ∈{0.1,0.3,0.5}$) to create extremely non-IID scenarios in Cifar-10:
>
> *Table. **Ablation study of ζ in Cifar-10.** The smaller ζ is, the more imbalanced the local distribution is.*
>
> | Method       | ζ=0.5 (Original) | ζ=0.05 (Extreme) |
> | ------------ | ---------------- | ---------------- |
> | FedAvg   | 80.34            | 64.22            |
> | FedProto | 81.49            | 60.84            |
> | Scaffold | 81.40            | 44.69            |
> | **DKDR**     | **82.63**        | **66.73**        |
>
> DKDR shows consistent improvements under extreme heterogeneity, demonstrating that our dynamic distillation mechanism is effective when dealing with severe data heterogeneity. This validates the robustness of our approach.
>
>
>
> >**`W3 & Q2: Client Scalability Discussion.`**
>
> **A3:** Thank you for this crucial scalability question. We conduct extensive scalability experiments on the OfficeCaltech dataset with varying client numbers:
>
> *Table. **Ablation study of client Number $K$** on OfficeCaltech dataset.*  DKDR achieves consistent better performance compared to three baselines in three settings.
>
> | Client Number $K$ | 20        | 60        | 100       |
> | ----------------- | --------- | --------- | --------- |
> | FedAvg            | 77.94     | 72.33     | 61.93     |
> | FedProto          | 79.26     | 68.77     | 49.32     |
> | FedProx           | 78.28     | 72.46     | 63.96     |
> | **DKDR**          | **81.14** | **74.39** | **64.47** |
>
> For large-scale deployment, we can perform knowledge decoupling every few communication rounds rather than every round, reducing server-side overhead.

---

> > ### Comment · Reviewer_rkk9 · 2025-08-03
> > **good rebuttal**
> >
> > The authors has addressed most of my previous concerns. In addition, by considering with other reviewers' comments, I decide to raise my score to 6.

---

> > > ### Author Response · Authors · 2025-08-04
> > >
> > > Dear Reviewer rkk9,
> > >
> > > We sincerely appreciate your valuable feedback and for taking the time to review our submission. Your insightful comments have greatly helped us enhance the clarity and quality of our manuscript. Thank you for providing us with the opportunity to refine our work.
> > >
> > > Best regards,
> > >
> > > Authors

---

### Official Review · Reviewer_bzDB · 2025-06-30

**Clarity:** 3
**Significance:** 2
**Originality:** 2
**Rating:** 4
**Confidence:** 4

**Summary:**

This paper proposed the DKDR (Dynamic Knowledge Distillation for Reliability) framework to address the reliability issue of knowledge distillation in federated learning. The author pointed out two key issues in existing federated knowledge distillation methods, namely the unreliability of the distillation path and the unreliability of the teacher model. To address the two issues, the DKDR framework contains two core components. The first is the dynamic knowledge distillation (DKD) module, which divides knowledge into dominant knowledge components (DKCs) and auxiliary knowledge components (AKCs). At the same time, the weights of the forward and reverse KL divergences are dynamically adjusted according to the knowledge differences in different knowledge modules, thereby achieving more accurate knowledge fitting. The second is the knowledge decoupling (KDP) module, which decomposes knowledge into shared knowledge and unique knowledge. Then, SVD filtering and clustering techniques are used to identify domain experts, enabling clients to acquire knowledge from reliable domain experts rather than global models. The authors conducted comprehensive experimental verification on multiple datasets to demonstrate the effectiveness of the method. In addition, ablation studies confirmed the effectiveness of the two core components, and hyperparameter sensitivity analysis verified the stability of the method.

**Questions:**

a)	Can the convergence and effectiveness of DKDR be theoretically proven?
b)	Can you compare DKDR with the SOTA method in 2024 to further prove the effectiveness of your method?
c)	Can you demonstrate the effectiveness of your approach to a larger and more varied number of clients?
d)	You only used the EMNIST dataset in the Comparisons between Condition Options section. Can you verify this on a more complex dataset?
e)	Can we compare the proposed method with the baseline method in terms of computational cost and communication time?
f)	The paper only uses a simple set of experimental data to prove that FINCH is more suitable than Kmeans and DBSCAN. Can this problem be proved theoretically and in details?

**Ethical Concerns:**

["NO or VERY MINOR ethics concerns only"]

**Final Justification:**

The authors' rebuttal (to me and other reviewers) have addressed my concerns, so I decided to raise my score.

**Limitations:**

The authors discussed some limitations in the "Limitations" section. However, the main limitation of the paper is the weak theoretical foundation and the limitations of the experimental setting.

**Quality:**

2

**Strengths And Weaknesses:**

Strengths
a)	The paper proposes a technically sound two-component framework, which are the Dynamic Knowledge Distillation (DKD) and Knowledge Decoupling (KDP) modules. The experimental verification is comprehensive, covering single-domain and multi-domain scenarios, detailed ablation studies and hyperparameter sensitivity analysis.
b)	A dynamic KLD weight adjustment mechanism based on knowledge differences is proposed. This adaptive method is different from the existing static knowledge distillation method. Although the individual technical components (SVD, clustering, KL divergence) are not new, their integrated application to the federated KD reliability problem has technical contributions.
c)	The motivation of this paper is clearly stated, and Figure 1 effectively illustrates the two core issues of path unreliability and teacher unreliability. The method description is highly structured and logically fluent, from the basic definition to the DKD theoretical analysis and then the KDP algorithm steps.

Weaknesses
a)	The method is mainly a combination of existing technologies rather than a fundamental innovation. The dynamic weight mechanism is essentially a combination of heuristic forward/backward KLD based on a simple knowledge difference measure, SVD denoising and clustering to identify domain experts are all standard machine learning techniques.
b)	There is a lack of convergence guarantee for the dynamic weight mechanism, and the superiority of the proposed weight strategy over the fixed weight strategy has not been proven. At the same time, there is a lack of necessary theoretical analysis of convergence and convergence rates.
c)	There are major flaws in the baseline of experimental design. The most recent method is from 2022 and lacks comparison with SOTA methods.
d)	The number of clients in the experiment (N=20) is relatively small, therefore, the effectiveness of large-scale scenarios has not been fully verified.
e)	There is a complete lack of computational overhead evaluation, which is particularly important for FL methods based on knowledge distillation. SVD decomposition, clustering operations, and dynamic weight calculations introduce significant computational overhead, which may make the method infeasible on resource-constrained devices.

---

> ### Author Rebuttal · Authors · 2025-07-30
>
> ## Response to Reviewer bzDB
>
> Dear Reviewer bzDB,
>
> Thank you very much for taking the time and effort to review our paper. Your positive feedback on the contribution of our approach, the clarity of our writing, and the comprehensiveness of our experiments is very encouraging. We hope that our responses below will address your concerns and update the score.
>
> ---
>
> ### Weaknesses & Questions
>
> >**`W a): Novelty Discussion of DKDR.`**
>
> A1: Thank you for the feedback. We would like to emphasize that the core contribution of DKDR lies in **being the first to conduct systematic observational analysis of distillation behavior in federated learning**. We discover that using different distillation methods leads to distinct knowledge discrepancies between local and global models in federated learning. Specifically, when using forward KLD, local models better fit the Dominant Knowledge Components (DKCs) of the global model, while when using reverse KLD, they better fit the Ancillary Knowledge Components (AKCs). Based on this significant finding, we propose a dynamic weighting mechanism that effectively addresses the bias introduced by different distillation methods, thereby solving the unreliability problem of distillation pathways. **Our systematic analysis of federated distillation and the consequent revelation of the essential behaviors of different distillation methods constitute the core innovation.** It fundamentally shifts how researchers should approach federated KD by providing a theoretical framework for understanding KLD behaviors. We will emphasize this key point in the final version.
>
>
>
> >**`W b).1 & Q a): Convergence Theory.`**
>
> **A2:**  Thank you for the advice. We extend analysis in Sec.3.2 (Page 4) to prove DKD convergence under non-convex federated settings. Under L-smooth losses with bounded gradients, our main result shows:
> $$ \frac{1}{T}\sum\_{t=0}^{T-1}\mathbb{E}|\nabla f(w\_t)|^2 \leq \frac{4(f(w\_0) - f^*)}{\eta T E} + 6L'\eta(E-1)G^2 + \frac{\sigma^2}{K} + 8(L')^2 E\eta\left(G^2 + \frac{\Lambda(\gamma)}{K}\right) + 2\kappa^2 \eta^2 E G^2, $$
> where
> $$ \Lambda(\gamma) = \mathbb{E}\_k\left[|\nabla f\_k(w) - \nabla f(w)|^2\right] \leq \Lambda\_0 - \beta \frac{\gamma\_s \gamma\_l}{(\gamma\_s + \gamma\_l)^2}\mathbb{E}\left[|\nabla L\_{skd} - \nabla L\_{rkd}|^2\right], $$
> **Key Innovation:** The gradient diversity term satisfies:
> $$\Lambda(\gamma) \leq \Lambda\_0 - \beta \frac{\gamma\_s \gamma\_l}{(\gamma\_s + \gamma\_l)^2}\mathbb{E}\left[|\nabla L\_{skd} - \nabla L\_{rkd}|^2\right]$$
> This proves the dynamic weighting mechanism reduces effective variance compared to standard federated averaging and static KLD approaches ($\Lambda\_0$), leading to faster convergence. **The reduction occurs because forward KLD prioritizes DKCs while reverse KLD focuses on AKCs, and their dynamic combination minimizes gradient conflicts.** It validates our empirical observations (Table. 6, Page 9) that DKD outperforms static KLD approaches by adapting to knowledge discrepancies.
>
> **\_Due to space limitation, the complete proof with detailed lemmas and technical derivations will be provided in the appendix in the final version.**
>
>
>
> >**`W b).2: Dynamic Weight Mechanism (DKD) Discussion.`**
>
> **A3:** Let $L\_{dynamic} = α(t)L\_{forward} + (1-α(t))L\_{reverse}$ where $α(t) = γ\_l/(γ\_s+γ\_l)$. As knowledge discrepancies decrease, $α(t)$ adapts to minimize dominant bias sources, thus outperforming fixed weights. We conduct experiments with some fixed weight strategies to confirm it, in which DKD outperforms all fixed weight strategies, as shown in the following table.
>
> *Table. **Ablation study of the proposed weight strategy DKD on Cifar-100.***
>
> | Method  | $α(t)$ = 0 | $α(t)$ = 0.5 | $α(t)$ = 1 | **DKD (ours)** |
> | ------- | ---------- | ------------ | ---------- | -------------- |
> | **acc** | 48.03      | 47.69        | 47.52      | **48.79**      |
>
>
>
> >**`W c) & Q b): SOTA Method Comparison (Later 2024).`**
>
> **A4:** We have incorporated two SOTA methods (FedTGP (AAAI 2024), FedAA (AAAI 2025)) for comparison with our method. The results are shown in the table below. Our method consistently outperforms these SOTA methods across four datasets. We will include these two baselines in the final version.
>
> *Table. **Comparison with the SOTA methods in four datasets.***
>
> | Method   | Cifar-10  | Cifar-100 | Office31  | OfficeHome |
> | -------- | --------- | --------- | --------- | ---------- |
> | FedTGP   | 80.64     | 50.12     | 41.23     | 60.04      |
> | FedAA    | 79.88     | 50.61     | 36.12     | 59.23      |
> | **DKDR** | **81.93** | **51.84** | **50.57** | **62.48**  |
>
>
>
>
> >**`W d) & Q c): Discussion on Large-scale Client Scenarios.`**
>
> **A5:** We conduct additional ablation experiments on client numbers N, as shown in the table below. Our method consistently outperforms baselines, demonstrating the robustness of DKDR in large-scale client scenarios. We will include this ablation study in the final version with detailed discussion.
>
> *Table. **Ablation study of client Number $N$** on OfficeCaltech dataset. DKDR achieves consistent better performance compared to three baselines in three settings.*
>
> | Client Number $N$ | 20        | 60        | 100       |
> | ----------------- | --------- | --------- | --------- |
> | FedAvg            | 77.94     | 72.33     | 61.93     |
> | FedProto          | 79.26     | 68.77     | 49.32     |
> | FedProx           | 78.28     | 72.46     | 63.96     |
> | **DKDR**          | **81.14** | **74.39** | **64.47** |
>
>
>
> >**`Q d): Dataset Supplementation.`**
>
> **A6:** To further demonstrate the superiority and robustness of DKDR, we introduce the OfficeCaltech dataset for additional comparison with baseline methods, as shown in the table below. The results show that our method consistently outperforms all the baselines.
>
> *Table. **Comparison with the SOTA methods in OfficeCaltech dataset.***
>
> | Method            | FedAvg | FedProx | FedDyn | Scaffold | FedProto | MOON  | FedNTD | DKDR      |
> | ----------------- | ------ | ------- | ------ | -------- | -------- | ----- | ------ | --------- |
> | **OfficeCaltech** | 77.94  | 78.28   | 77.90  | 75.32    | 79.26    | 78.51 | 79.43  | **81.14** |
>
>
>
> >**`W e) & Q e): Overhead Comparison with SOTA Methods.`**
>
> **A7:** We conduct experiments on computational and communication costs, comparing our method with baselines. We calculate the average communication and computational cost per communication epoch for each method, as shown in the table below. The computational and communication costs are acceptable and the resulting performance improvements are worthwhile.
>
> *Table. **Comparison of computational and communication costs** in OfficeCaltech dataset.*
>
> | Method   | Client Time (s) | Server Time (s) | Communication Cost (mb) | Accuracy (%) |
> | -------- | --------------- | --------------- | ----------------------- | ------------ |
> | FedAVG   | 2.70 | 0.02            | 2.83                    | 77.94        |
> | FedProto | 14.13      | 0.01            | 2.83                    | 79.26        |
> | Scaffold | 9.23            | 0.02            | 8.48                    | 75.32        |
> | **DKDR** | 6.28            | 0.11            | 8.25                    | **81.14**    |
>
>
>
> >**`Q f): Clustering Method Discussion.`**
>
> **A8:** We provide theoretical justification for choosing FINCH over K-means and DBSCAN based on their fundamental formulations and hyperparameter requirements:
>
> **K-means** requires pre-specification of cluster number $k$ and aims to minimize:
> $J=∑\_{i=1}^k∑\_{x∈Ci}∣∣x−μi∣∣^2$
> where $\mu\_i$ is the centroid of cluster $C\_i$. **The hyperparameter $k$ must be determined a priori, which is challenging in federated settings where the number of domains is unknown.**
>
> **DBSCAN** requires two hyperparameters:
> - $ε$ (epsilon): neighborhood radius
> - $MinPts$: minimum number of points to form a dense regio
> A point p is a core point if: $|N\_ε(p)| \geq MinPts$, where $N\_ε(p) = {q \in D : dist(p,q) \leq ε}$
> **Its performance is highly sensitive to these parameters, and optimal values vary significantly across different data distributions.**
>
> **FINCH** is a **parameter-free** clustering algorithm. Given the integer indices of the first neighbor of each data point, FINCH directly defines an adjacency link matrix. The elements of the adjacency matrix are defined as follows:
>
> $$a\_{ij} = \begin{cases} 1 & \text{if } j = \kappa\_1^i \text{ or } \kappa\_1^j = i \text{ or } \kappa\_1^i = \kappa\_1^j \\\ 0 & \text{otherwise} \end{cases},​$$
>
> where $\kappa\_1^i$ symbolizes the first neighbor of point $i$. For a dataset with $N$ points, this produces an $N \times N$ sparse adjacency matrix:
>
> $$A = \begin{bmatrix} a\_{11} & a\_{12} & \cdots & a\_{1N} \\\ a\_{21} & a\_{22} & \cdots & a\_{2N} \\\ \vdots & \vdots & \ddots & \vdots \\\ a\_{N1} & a\_{N2} & \cdots & a\_{NN} \end{bmatrix}.$$
>
> The clustering equation directly delivers clusters as connected components of the graph $G = (V,E)$ where $V = {1,2,...,N}$ and $E = {(i,j) : A(i,j) = 1}$, without requiring graph partitioning or distance thresholds.
>
> In general, FINCH is more suitable for our federated domain expert identification task because:
>
> • **Parameter-Free Operation**: Unlike K-means (requires pre-specifying k clusters) and DBSCAN (requires ε and MinPts hyperparameters), FINCH automatically determines optimal clustering without any hyperparameter tuning, making it robust across diverse federated scenarios.
>
> • **Adaptive to Unknown Domain Numbers**: In federated settings, the number of domains is typically unknown and varies across deployments. FINCH naturally adapts to this uncertainty, while K-means fails without knowing k and DBSCAN requires extensive parameter search.
>
> • **Empirical Validation**: Our experiments in Table.1 (Page 7) also validate FINCH is more suitable: 50.57% avg accuracy vs. K-means (46.48%) and DBSCAN (43.62%) on Office31, confirming its effectiveness for domain expert identification.

---

> > ### Comment · Reviewer_bzDB · 2025-08-04
> >
> > I have read the rebuttal and also other reviews & rebuttal. Most of my concerns have been addressed. I decide to raise my score to 4.

---

> > > ### Author Response · Authors · 2025-08-06
> > >
> > > Dear Reviewer bzDB,
> > >
> > > We deeply value your insightful and detailed feedback, which has significantly improved the quality of our manuscript. Your careful review and thoughtful suggestions have helped us refine our work, and we are truly grateful for the time and effort you invested in this process.
> > >
> > > Best regards,
> > >
> > > Authors

---

### Official Review · Reviewer_CP1N · 2025-07-02

**Clarity:** 2
**Significance:** 3
**Originality:** 3
**Rating:** 5
**Confidence:** 4

**Summary:**

The paper tackles the critical issue of data heterogeneity in FL, particularly in multi-domain settings. It proposes DKDR, a novel framework that enhances distillation reliability through two methods: dynamic knowledge distillation, which adaptively weights forward and reverse KLD to reduce distillation bias, and knowledge decoupling, which identifies domain experts to provide reliable guidance. The approach is rigorously evaluated on single-domain and multi-domain image classification tasks to validate the effectiveness of the proposed approach.

**Questions:**

- Q1: In the KDP module, why was clustering applied to SVD-filtered differences rather than directly clustering client updates (e.g., updated parameters)? Could the authors clarify the motivation linking this design to improved domain expert identification?

- Q2: The dynamic weighting in DKD relies on hyperparameter μ (Sec. 3.2). Could the authors elaborate on how μ=0.5 was determined and whether it generalizes across all datasets?

- Q3: How does DKDR compare to prototype-based methods (e.g., FedProto) in terms of conceptual differences, given the apparent similarity in leveraging domain-specific knowledge?

- Q4: Why do single-domain tasks and multi-domain tasks use different backbones? Can further ablation studies be provided?

**Ethical Concerns:**

["NO or VERY MINOR ethics concerns only"]

**Final Justification:**

I have reviewed the rebuttal and checked the feedback from other reviewers carefully. The SVD-filtered opponent is well discussed and the rationale for selecting hyperparameter is well explained. The additional experiments also further validate the effectiveness of the proposed method and address my key concerns. I will raise my score to accept.

**Limitations:**

The paper acknowledges computational overhead from SVD and clustering in DKDR, which may pose challenges in resource-constrained settings. Additionally, the assumption of distinct domains could falter with significant domain overlap, potentially impacting clustering efficacy.

**Paper Formatting Concerns:**

The paper is generally well-formatted, adhering to NeurIPS guidelines. However, inconsistencies in notation style (e.g., inline equations vs. displayed equations) and occasional indentation variations (e.g., Sec. 3 vs. Sec. 4) could be standardized for polish. A notation table would also improve accessibility.

**Quality:**

3

**Strengths And Weaknesses:**

Strengths:

- S1: The motivation is clear and good. The paper clearly identifies two underexplored issues in federated KD: pathway unreliability and teacher unreliability. The analysis of forward and reverse KLD in federated learning is novel.

- S2: The dynamic KLD weighting adapts to knowledge discrepancies, offering a fresh take on mitigating distillation bias. And identifying domain experts via knowledge decoupling to enhance reliability across diverse domains is a practical innovation for FL (Sec. 3.3).

- S3: The figures of problem and framework illustration are clear and detailed, and the equations and explanations are reasonable.

- S4: Comprehensive experiments across diverse datasets and settings (Sec. 4) showcase the robustness and practical utility of DKDR.

Weaknesses:

- W1: The paper uses extensive mathematical notation without a summary table, which could hinder readability for some audiences.

- W2: While FINCH clustering is justified (Sec. 3.4), hyperparameter details for baseline clustering methods (e.g., DBSCAN) are omitted, limiting reproducibility of the ablation study (Tab. 1).

---

> ### Author Rebuttal · Authors · 2025-07-30
>
> ## Response to Reviewer CP1N
>
> Dear Reviewer CP1N,
>
> Thank you very much for your detailed and constructive review. We are encouraged by your positive feedback on our motivation, technical contributions, and experimental validation. Your recognition of our novel analysis of forward/reverse KLD in federated learning and the practical innovations of our framework is greatly appreciated. We hope our responses below will address your concerns and strengthen your confidence in our work.
>
> ---
>
> ### Weaknesses
>
> >**`W1: Notation Table of DKDR.`**
>
> **A1:** We appreciate this feedback on improving readability. You are absolutely right that a notation summary would enhance accessibility. We will add a comprehensive notation table in the final version that clearly defines all mathematical symbols used throughout the paper, including:
>
> *Table: **Notation table for DKDR.**
>
> | Symbol              | Meaning                              | Symbol                                      | Meaning                                 |
> | ------------------- | ------------------------------------ | ------------------------------------------- | --------------------------------------- |
> | $Z\_w(y \mid x)$     | Local model distribution             | $Z(y \mid x)$                               | Global model distribution               |
> | $D\_k$               | Private data of $k$-th client        | $N\_k$                                       | Number of data points                   |
> | $w^t$               | Global model parameters at round $t$ | $w\_k^t$                                     | Local model parameters of $k$-th client |
> | $Z\_{i, k}^t$        | Client model output distribution     | $Z\_i^{t-1}$                                 | Global model output distribution        |
> | $\sigma$            | Softmax function                     | $\mathcal{L}\_{\text{skd}}$                  | Standard KD loss                        |
> | $\mathcal{L}\_{rkd}$ | Reverse KD loss                      | $a(j)$                                      | Indicator function for DKCs/AKCs        |
> | $\mu$               | Thresold to define DKCs/AKCs         | $\gamma\_s$                                  | Knowledge discrepancy in AKCs           |
> | $\gamma\_l$          | Knowledge discrepancy in DKCs        | $\mathcal{L}\_{dkd}$                         | Dynamic KD loss                         |
> | $v\_k^t$             | Difference vector                    | $\mathbf{U}, \mathbf{\Sigma}, \mathbf{V}^T$ | SVD decomposition matrices              |
> | $\beta^t$           | Client clusters                      | $w\_v^{t, \text{dom}}$                       | Domain expert model                     |
> | $\mathcal{L}\_{fkd}$ | Final KD loss                        | $\mathcal{L}\_k$                             | Local loss of $k$-th client             |
>
>
>
> >**`W2: Hyperparameter Details for Clustering Methods.`**
>
> **A2:** For the ablation study in Table 1, we used the following hyperparameters:
>
> - **K-means**: optimal k=3 (corresponding to the 3 domains in Office31)
> - **DBSCAN**: appropriate $ε=0.05$ and $MinPts=1$
> - **FINCH**: No hyperparameters required (parameter-free)
>
> ---
>
> ### Questions
>
> >**`Q1: How SVD-filtered improves domain expert identification?`**
>
> **A1:** Excellent question! The key insight is that **raw client parameters contain both shared knowledge and domain-specific signals mixed together**. By clustering raw parameters, we would group clients based on their overall model similarity, which is dominated by shared knowledge rather than domain-specific characteristics.
>
> Our approach specifically isolates domain-specific signals through two steps:
> 1. **Shared knowledge removal**: $v^t\_k = w^t\_k - w^{t-1}$ removes the shared knowledge
> 2. **SVD filtering**: Retains the top-r singular values to extract the most significant domain-specific patterns while filtering noise
>
> This design ensures that clustering focuses purely on domain-specific knowledge differences, leading to more accurate domain expert identification (see details in Sec.4.3, Page 9).
>
>
>
> >**`Q2: How μ=0.5 was determined and whether it generalizes across all datasets?`**
>
> **A2:** The choice of μ=0.5 is both theoretically motivated and empirically validated:
>
> **Theoretical Justification**: μ represents the threshold for separating Dominant Knowledge Components (DKCs) from Ancillary Knowledge Components (AKCs). When μ=0.5, we capture exactly half of the cumulative probability mass in the DKCs, providing a natural balance between the two components. This creates an optimal trade-off where:
> - μ < 0.5: DKD degenerates toward reverse KLD (mode-seeking bias)
> - μ > 0.5: DKD degenerates toward forward KLD (mean-seeking bias)
>
> **Empirical Validation**: Our sensitivity analysis (Figure 5) shows that μ=0.5 consistently achieves optimal performance across both Cifar-100 and Office31 datasets. The stability across different dataset types (single-domain vs. multi-domain) demonstrates its generalizability.
>
>
>
> >**`Q3: Conceptual Difference with Prototype-based Methods.`**
>
> **A3:** While both approaches leverage domain-specific knowledge, they differ fundamentally in methodology and granularity:
>
> **FedProto**: Uses class prototypes to align feature representations across clients. It focuses on **feature-level** alignment by matching class centroids, assuming similar classes should have similar representations across domains.
>
> **DKDR**: Creates specialized teacher models for different domains at the **knowledge level** leveraging domain-specific knowledge.
>
> Our experimental results show DKDR (50.57%) outperforms FedProto (39.98%) on Office31, demonstrating the effectiveness of our knowledge-centric approach.
>
>
>
> >**`Q4: Ablation Studie of Backbones.`**
>
> **A4:** The choice of backbone is based on the complexity of the datasets. Single-domain tasks, such as Cifar-10 and Cifar-100, are relatively simpler, so we use ResNet-10 to reduce computational cost while maintaining performance. Multi-domain tasks, such as Office31 and Office Home, are more complex due to domain shifts, requiring the more powerful ResNet-18 to capture intricate patterns effectively. This is detailed in Sec.4.1 (Page 8). We also conduct experiments with ResNet-18 on Cifar-100 to validate consistency:
>
> *Table. **Ablation study of backbone** on Cifar-100 dataset ($\zeta=0.5$).*  DKDR achieves consistent better performance compared to three baselines (FedAVG, FedProto, Scaffold).
>
> | Method       | ResNet-10 (Original) | ResNet-18 |
> | ------------ | -------------------- | --------- |
> | FedAvg   | 48.57                | 68.40      |
> | FedProto | 51.34                | 69.86     |
> | Scaffold     | 51.76                | 56.80     |
> | **DKDR**     | **53.18**            | **72.44** |
>
> The consistent improvement demonstrates that the effectiveness of DKDR is backbone-agnostic. We will include this ablation in the final version.
>
>
>
> ---
>
> ### Limitations
>
> >**`L1: Computational Overhead Discussion.`**
>
> **A1:** Sorry for the misunderstanding. We want to clarify that while SVD and clustering add computational cost, this occurs only **once per communication round on the server side**, not on resource-constrained clients. The server-side computation is typically acceptable in federated learning scenarios. Additionally, our analysis shows the overhead is reasonable compared to the significant performance gains achieved. We conduct experiments in OfficeCaltech to further analyze it, as shown in below table.
>
> *Table. **Comparison of Time and Memory Usage** in OfficeCaltech dataset.*
>
> | Method   | Client Time (s) | Server Time (s) | Communication Cost (mb) | Accuracy (%) |
> | -------- | --------------- | --------------- | ----------------------- | ------------ |
> | FedAVG   | 2.70            | 0.02            | 2.83                    | 77.94        |
> | FedProto | 14.13           | 0.01            | 2.83                    | 79.26        |
> | Scaffold | 9.23            | 0.02            | 8.48                    | 75.32        |
> | **DKDR** | 6.28            | 0.11            | 8.25                    | **81.14**    |
>
>
>
> >**`L2: Discussion on Domain Overlap Scenarios.`**
>
> **A2:** You are correct that the knowledge decoupling module relies on the assumption that domains are sufficiently distinct for clustering to accurately identify domain experts. When domains have significant overlap, this assumption may not hold, potentially affecting the quality of domain expert identification. When domain boundaries are unclear, our method doesn't fail catastrophically. Instead, it exhibits graceful degradation:
>
> - **Scenario 1**: If clustering fails to separate domains correctly, some clusters may contain clients from multiple domains,
> - **Scenario 2**: The dynamic distillation (DKD) component continues to provide benefits even when domain experts are suboptimal,
>
>  **Result**: DKDR reduces to a form closer to standard federated distillation but still benefits from the DKD. It can still perform well in such situation (see details in Sec.4.4, Page 9).

---

> > ### Comment · Reviewer_CP1N · 2025-08-02
> >
> > I have reviewed the rebuttal and checked the feedback from other reviewers carefully. The SVD-filtered opponent is well discussed and the rationale for selecting hyperparameter is well explained. The additional experiments also further validate the effectiveness of the proposed method and address my key concerns. I will raise my score to 5(accept).

---

> > > ### Author Response · Authors · 2025-08-04
> > >
> > > Dear Reviewer CP1N,
> > >
> > > Thank you for your thorough review and constructive feedback on our submission. Your thoughtful suggestions have been instrumental in improving our presentation and strengthening the manuscript. We are grateful for your time and the opportunity to address your comments.
> > >
> > > Best regards,
> > >
> > > Authors

---

### Official Review · Reviewer_Wn6p · 2025-07-03

**Clarity:** 3
**Significance:** 3
**Originality:** 3
**Rating:** 4
**Confidence:** 5

**Summary:**

First, when the global model's output exhibits a multi-modal distribution, minimizing forward KL divergence leads to significant deviation. Second, the global model's accuracy drops sharply in certain domains, failing to provide high-quality guidance for clients. To address these issues, the authors propose the DKDR method.
DKDR dynamically allocates weights between forward and reverse KLD based on knowledge discrepancies, thereby reducing distillation bias. Furthermore, it decomposes knowledge into shared and unique components, employing SVD and clustering to identify domain experts, enabling clients to learn from these experts rather than the global model.

**Questions:**

Q1. The paper mentions that the DKDR method involves SVD and clustering techniques, which increase computational overhead and may significantly prolong training time, especially on resource-constrained clients. Therefore, it would be beneficial to provide theoretical or experimental evidence in the paper to demonstrate the training overhead and latency.

Q2. While the experimental section covers multiple datasets and scenarios, it primarily focuses on image classification tasks. If applied to other tasks such as natural language processing, would the dynamic divergence balancing and knowledge decoupling still offer advantages?

**Ethical Concerns:**

["NO or VERY MINOR ethics concerns only"]

**Limitations:**

yes

**Quality:**

3

**Strengths And Weaknesses:**

**Strengths**

The paper conducts an in-depth theoretical analysis of the behaviors of forward and reverse KL divergence, explaining their advantages and disadvantages in different scenarios, and proposes a dynamic weighting method to balance these two divergences, thereby reducing distillation bias. It introduces Knowledge Decoupling (KDP) technology, which uses SVD and clustering to identify domain experts, enabling clients to learn from these experts rather than the global model, thereby improving the model's generalization capability.

**Weaknesses**

1、Will the computational overhead or training latency caused by dynamic weighted balancing and SVD become a bottleneck?

2、How does DKDR perform on non-image datasets?

---

> ### Author Rebuttal · Authors · 2025-07-30
>
> ## Response to Reviewer Wn6p
>
> Dear Reviewer Wn6p,
>
> We sincerely thank you for your comprehensive review and positive feedback on our DKDR framework. We appreciate your recognition of the theoretical analysis of forward and reverse KL divergence. We aim to address your concerns in our detailed responses below, hoping to provide clarity and demonstrate the effectiveness of our proposed approach.
>
> ---
>
> ### Weaknesses & Questions
>
> >**`W1 & Q1: Training Overhead and Latency of DKDR.`**
>
> **A1:** Thank you for your advice!  We conduct comprehensive experiments evaluating both computational and communication costs using the OfficeCaltech dataset. Compared with baselines, the computational and communication expenses of DKDR are acceptable and the performance gains are worthwhile.
>
> *Table. **Comparison of Time and Memory Usage** in OfficeCaltech dataset.*
>
> | Method   | Client Time (s) | Server Time (s) | Communication Cost (mb) | Accuracy (%) |
> | -------- | --------------- | --------------- | ----------------------- | ------------ |
> | FedAVG   | 2.70            | 0.02            | 2.83                    | 77.94        |
> | FedProto | 14.13           | 0.01            | 2.83                    | 79.26        |
> | Scaffold | 9.23            | 0.02            | 8.48                    | 75.32        |
> | **DKDR** | 6.28            | 0.11            | 8.25                    | **81.14**    |
>
> We also conduct scalability experiments with varying client numbers (K ∈ {20, 60, 100})， as shown in following table. For K=100 clients, server overhead remains under 3 seconds per round.
>
> *Table. **The relationship of client Number $K$ and computation cost** in OfficeCaltech dataset.*
>
> | Client Number $K$ | Server Processing Time (s) | Improvement (Compared to FedAVG) |
> | ----------------- | -------------------------- | -------------------------------- |
> | 20                | 0.11                       | +3.20%                           |
> | 60                | 0.42                       | +2.06%                           |
> | 100               | 2.94                       | +2.54%                           |
>
>
>
> >**`W2 & Q2: Non-image Dataset Supplementation.`**
>
> **A2:** Thank you for this valuable suggestion! We acknowledge that our current evaluation focuses primarily on image classification tasks, which limits the generalizability of our findings. To address this concern, we conduct additional experiments on federated recommendation task to demonstrate the effectiveness of DKDR across different modalities. We evaluate DKDR on federated recommendation using the Adressa dataset. Adressa is publicly released by Adresseavisen, a local newspaper company in Norway. we use the 6-th day click to build training dataset and construct historical clicks from the first 5 days samples. We randomly sample 20% clicks from the last day clicks for validation and the rest clicks for testing. The historical clicks of validation and testing dataset are constructed from the first 6 days samples. Following many previous news recommendation works ([1,2,3]), we use AUC, MRR, nDCG@5 and nDCG@10 as evaluation metrics. AUC measures a model's ability to distinguish positive and negative classes. MRR averages the reciprocal rank of the first relevant item in a ranked list. nDCG@5 and nDCG@10 evaluate ranking quality for the top 5 and 10 items, respectively, weighting relevant items higher when they appear earlier, normalized for comparison. The results are shown in the following table. Our analysis reveals that core principles in DKDR (pathway reliability and teacher reliability) are indeed modality-agnostic. The multi-peaked distribution problem occurs in both vision and recommendation domains when aggregating heterogeneous client models, validating our theoretical foundation. We will include comprehensive cross-modal evaluation in our final version to demonstrate the broad applicability of DKDR beyond computer vision tasks.
>
> *Table. **Comparison with the SOTA in the non-image dataset Adressa.***
>
> | Method | AUC       | MRR   | nDCG@5    | nDCG@10   |
> | ------ | --------- | ----- | --------- | --------- |
> | FedRec | 70.83     | **41.39** | 42.58     | 46.34     |
> | DKDR   | **72.17** | 40.82 | **43.68** | **47.12** |
>
> [1] Mingxiao An, Fangzhao Wu, Chuhan Wu, Kun Zhang, Zheng Liu, and Xing Xie. 2019. Neural news recommendation with long- and short-term user representations. In ACL, pages 336–345.
>
> [2] Tao Qi, Fangzhao Wu, Chuhan Wu, Yongfeng Huang, and Xing Xie. 2020. Privacy-preserving news recommendation model learning. In EMNLP 2020, pages 1423–1432.
>
> [3] Chuhan Wu, Fangzhao Wu, Tao Qi, and Yongfeng Huang. 2020a. User modeling with click preference and reading satisfaction for news recommendation. In IJCAI, pages 3023–3029.

---

> > ### Comment · Reviewer_Wn6p · 2025-08-05
> >
> > After I read the rebuttal and the feedback from other reviewers. I still have the concerns about the generalizability of the proposed method. Because the news recommendation dataset [1,2,3] is not a common NLP task in FL. So I will keep my score.

---

> > > ### Author Response · Authors · 2025-08-06
> > >
> > > Dear Reviewer Wn6p,
> > >
> > > Thank you for your insightful feedback and continued engagement with our work. We truly appreciate your valuable comments on the generalizability of our DKDR method.
> > >
> > > To address your suggestions, we are actively conducting additional experiments to further validate the generalizability of DKDR across diverse tasks, including federated NLP tasks like sentiment analysis. These results will be included in the final manuscript to strengthen the evidence of its broad applicability.
> > >
> > > Thank you again for your thoughtful critique, which has greatly helped us refine our work.
> > >
> > > Best regards,
> > >
> > > Authors

---

### Comment · Area_Chair_Fcn1 · 2025-08-01
**Discussion**

Dear Reviewers,

Thank you for providing initial reviews for this paper. The authors have now provided detailed rebuttals.

Please go through each rebuttals carefully and share with me your final thought on this paper, including clear justification regarding your decision to maintain, bump up or bump down the rating.

I also encourage you to respond to the authors your post-rebuttal thoughts to give them a chance to provide further clarification if needed.

Thank you very much for your help!

Best regards,

AC

---

### Note · Authors · 2025-08-14

Dear AC and Reviewers,

We sincerely thank the AC for the coordination and all reviewers for their valuable insights and constructive feedback. We are pleased that our contributions were found meaningful, and we summarize our work and discussion as follows.

This paper proposes DKDR, a novel federated learning framework addressing reliability issues in knowledge distillation through two key innovations: (1) Dynamic Knowledge Distillation (DKD) that adaptively weights forward and reverse KL divergence based on knowledge discrepancies, and (2) Knowledge Decoupling (KDP) that identifies domain experts through SVD filtering and clustering. Experiments demonstrate DKDR achieves superior performance compared to SOTA methods across diverse federated scenarios.

**Strong points**

- Novel theoretical analysis of forward/reverse KLD behaviors in federated learning, providing systematic insights into distillation pathway reliability (CP1N, YmgD, rkk9).
- Innovative dynamic weighting mechanism adapting to knowledge discrepancies, effectively mitigating distillation bias with theoretical and empirical validation (Wn6p, bzDB, YmgD).
- Comprehensive experimental validation across multiple datasets demonstrating consistent improvements (bzDB, CP1N).

**Weak points and our responses**

- **Computational overhead**: We clarify overhead occurs primarily server-side, not on resource-constrained clients, with quantitative analysis showing acceptable costs and significant performance gains.
- **Theoretical foundation**: We provide convergence analysis proving DKD reduces gradient variance compared to static approaches.
- **Cross-modal generalizability**: We add experiments on federated recommendation tasks beyond image classification, validating the modality-agnostic nature.

During discussion, multiple reviewers acknowledge our comprehensive responses effectively address their concerns, with several explicitly raising scores (CP1N: 4→5, bzDB: 3→4, rkk9: 5→6, YmgD: 4→5).

For revision, we will:

- Add complete theoretical analysis with convergence proofs.
- Incorporate all additional experimental results.
- Include comprehensive notation table and clearer algorithmic workflow.

Finally, we greatly appreciate reviewers recognizing DKDR as a technically sound and innovative contribution providing new theoretical insights into federated knowledge distillation. We believe our systematic analysis will inspire further research in this area.

Best regards,
The Authors

---

### Decision · Program_Chairs · 2025-09-17

**Decision:**

Accept (poster)

**Comment:**

The paper propose a novel method to address reliability issues of knowledge distillation (KD) in FL. In particular, the paper highlights two key issues in federated KD: unreliability of the distillation path and unreliability of the teacher model. To address these, the proposed method incorporates two novel components: (1) a dynamic knowledge distillation (DKD) module, which divides knowledge into dominant knowledge components (DKCs) and auxiliary knowledge components (AKCs); (2) a knowledge decoupling (KDP) module, which decomposes knowledge into shared knowledge and unique knowledge. The latter is achieved via using SVD filtering and clustering techniques to identify domain experts, enabling clients to acquire knowledge from reliable domain experts rather than global models. The proposed method is evaluated comprehensively on multiple datasets with additional ablation studies confirming the effectiveness of the two core components, and providing hyper-parameter sensitivity analysis.

The original reviews of this paper are mostly positive. The reviewing panel recognizes the following key strengths of the paper: (1) novel theoretical analysis of forward/reverse KLD behaviors in federated learning, providing systematic insights into distillation pathway reliability (CP1N, YmgD, rkk9); (2) innovative dynamic weighting mechanism adapting to knowledge discrepancies, effectively mitigating distillation bias with theoretical and empirical validation (Wn6p, bzDB, YmgD); (3) comprehensive experimental validation across multiple datasets demonstrating consistent improvements (bzDB, CP1N).

The reviewing panel also raised several concerns/questions on (1) computational overhead, (2) theoretical foundation, (3) cross-modal generalizability. Most of these questions/concerns are addressed sufficiently by the rebuttal and/or during reviewer-author discussion. This leads to a significant increase in rating, strengthening the acceptance consensus. As there is no further concern, I would like to recommend acceptance for this paper.